# SoftJAX & SoftTorch:
# Empowering Automatic Differentiation Libraries with Informative Gradients

**Anselm Paulus** [* 1]   **A. René Geist** [* 1]   **Vít Musil** [2]   **Sebastian Hoffmann** [3]   **Georg Martius** [1 4]

## Abstract

Automatic differentiation (AD) frameworks such as JAX and PyTorch have enabled gradient-based optimization for a wide range of scientific fields. Yet, many "hard" primitives in these libraries such as thresholding, Boolean logic, discrete indexing, and sorting operations yield zero or undefined gradients that are not useful for optimization. While numerous "soft" relaxations have been proposed that provide informative gradients, the respective implementations are fragmented across projects, making them difficult to combine and compare. This work introduces SoftJAX and SoftTorch, open-source, feature-complete libraries for *soft differentiable programming*. These libraries provide a variety of soft functions as drop-in replacements for their hard JAX and PyTorch counterparts. This includes (i) elementwise operators such as $\mathrm{clip}$ or $\mathrm{abs}$, (ii) utility methods for manipulating Booleans and indices via fuzzy logic, (iii) axiswise operators such as $\mathrm{sort}$ or $\mathrm{rank}$ – based on optimal transport or permutahedron projections, and (iv) offer full support for straight-through gradient estimation. Overall, SoftJAX and SoftTorch make the toolbox of soft relaxations easily accessible to differentiable programming, as demonstrated through benchmarking and a practical case study. Code is available at `github.com/a-paulus/softjax` and `github.com/a-paulus/softtorch`.

## 1. Introduction

**Automatic differentiation (AD) frameworks have en-**

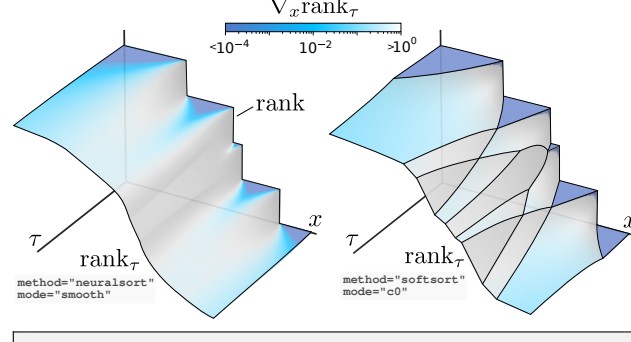

*Figure 1.* **Top:** SoftJAX and SoftTorch provide differentiable surrogates for discrete operations i. e., $\mathrm{rank}_\tau$ instead of $\mathrm{rank}$. Numerous soft approximations can be obtained by tuning the softness parameter $\tau$, selecting the softening method (e. g., "neuralsort" or "softsort"), and choosing a smoothness mode (e. g., `smooth` for $\mathcal{C}^\infty$ or `c1` for $\mathcal{C}^1$). **Bottom:** The soft surrogates shown above are instantiated using just three lines of code.

* Equal contribution. Libraries implemented by Anselm Paulus. [1]University of Tübingen, Germany [2]Masaryk University, Czechia [3]Max Planck Institute for Biogeochemistry, Germany [4]Max Planck Institute for Intelligent Systems, Tübingen, Germany. Correspondence to: Anselm Paulus <anselm.paulus@tuebingen.mpg.de>, A. René Geist <rene.geist@uni-tuebingen.de>.

*Proceedings of the 43rd International Conference on Machine Learning*, Seoul, South Korea. PMLR 306, 2026. Copyright 2026 by the author(s).

**abled rapid progress in machine learning.** They make gradient computation efficient, user-friendly, and composable. Thereby, they become a ubiquitous tool widely adopted beyond machine learning. This includes (i) *differentiable rendering* (Wang et al., 2019) using Nerfs (Mildenhall et al., 2020) and Gaussian splatting (Kerbl et al., 2023); (ii) *differentiable simulation*, such as MuJoCo XLA (MJX) (Todorov et al., 2012; MuJoCo, 2024; Paulus et al., 2026) and WARP (Howell et al., 2025); (iii) *structured prediction* like ranking (Rolínek et al., 2020a) and matching (Rolínek et al., 2020b); (iv) *combinatorial layers* for CEM (Amos & Yarats, 2020), MPC (Amos et al., 2018), and discrete decisions (Vlastelica et al., 2020; Berthet et al., 2020); (v) *differentiable optimization* (Amos & Kolter, 2017; Agrawal et al., 2019; Blondel et al., 2022); and (vi) *physical simulations*, e. g., for a gravitational wave detector (Ruiz-Gonzalez et al., 2025). However, the programs for these and many more applications contain classical operations using comparisons for branching code, ranking elements, and so forth.

**Unfortunately, AD does not necessarily result in informative gradients.** Here, *informative* means providing a stable local direction leading to improvements in the

objective. Uninformative are cases with zero gradients (e. g., rounding, comparisons, indexing, sort/max/top-k/median) and arbitrary subgradients (ReLU at zero, sort with duplicates). Many applications call for replacing such hard operations with soft ones that yield informative gradients. A successful example is addressing the "dying ReLU problem" (Lu et al., 2020), where ReLU's zero gradients hinder gradient-based optimization of deep neural networks, by replacing the ReLU function with a smooth relaxation such as SiLU or Softplus.

**While many relaxation techniques have been proposed, the existing toolbox is fragmented.** Proposed techniques include analytic smooth surrogates (e. g., SiLU for ReLU (Hendrycks & Gimpel, 2016), sigmoid for steps (Petersen et al., 2021; 2022b), and softmax for argmax (Beker et al., 2025; 2026)), optimization-based approaches (e. g., sorting via regularized optimal transport (Cuturi et al., 2019; Xie et al., 2020; Blondel et al., 2018) or projections onto the permutahedron (Blondel et al., 2020; Sander et al., 2023), proximal methods (Paulus et al., 2024)), stochastic relaxations (e. g., Gumbel-softmax (Maddison et al., 2017; Jang et al., 2017; Paulus et al., 2020), perturbed optimizers (Berthet et al., 2020; Niepert et al., 2021)), gradient "hacks" / estimators (straight-through (Bengio et al., 2013; Sahoo et al., 2023), black-box differentiation (Vlastelica et al., 2020)) and more.

**To unify soft relaxations and ease their use, we present** *SoftJAX* **and** *SoftTorch***.** Our libraries provide soft drop-in replacements for many commonly used hard operations in JAX (Bradbury et al., 2018) and Pytorch (Paszke et al., 2019). In turn, SoftJAX and SoftTorch form easy to use abstraction layers between user code and underlying primitive operations. Our treatment also draws unifying connections, generalizes various techniques, and provides comprehensible "softness" knobs and "mode" families across operations. As an example, see Figure 1 where we plot the soft rank operator for varying softness parameters $\tau > 0$ and softening modes. As for all our relaxations, we recover the original hard operation as $\tau \to 0^+$. By combining the vast toolbox of soft relaxations, our libraries make *soft differentiable programming* accessible.

We next explain the operators implemented in SoftJAX and SoftTorch and how their hard versions are relaxed to soft counterparts. Section 2 reviews the basics of soft surrogates and straight-through estimation. Section 3 then describes the softening of elementwise operators in SoftJAX and SoftTorch, such as sign or round, via relaxations of the Heaviside step function. Section 4 shows how more involved axiswise operators, such as sort or quantile, can be softened via optimal transport or projections onto the unit simplex or the permutahedron. An overview of all currently implemented functions in SoftJAX is available in Figure 9.

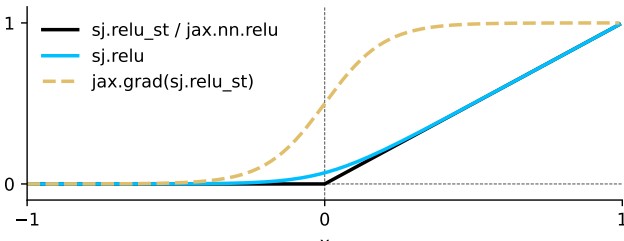

*Figure 2.* The function `relu_st` resorts to the straight-through trick to use the hard function `jax.nn.relu` in the forward pass while using the soft function `sj.relu` for gradient computation.

Finally, Section 5 presents runtime and memory comparisons to help end users select suitable methods that balance the trade-offs required by their applications.

## 2. Softening and Straight-Through Estimation

Many functions used in modern ML frameworks are poorly suited for automatic differentiation as they are discontinuous or have large regions with zero derivative. To resolve this, our libraries have one main goal: to provide informative gradients for any program written in the supported frameworks. The underlying mechanism rests on two core concepts: *soft surrogates* and *straight-through estimation*.

**Soft surrogate.** *Soft surrogates* replace the original function with another function that has more informative gradients for optimization. Specifically, we call the function $f_\tau$ with softening parameter $\tau > 0$ a *soft surrogate* of $f$ if (i) $f_\tau$ is continuous and differentiable almost everywhere, (ii) $f_\tau$ yields informative gradients over the relevant domain, i. e., avoids extended regions of zero derivative, and (iii) $f_\tau$ recovers $f$ in the limit $\tau \to 0^+$. The softening parameter $\tau$ controls the trade-off between faithfulness to $f$ and gradient informativeness: larger $\tau$ benefits differentiability, while $\tau \to 0^+$ causes $f_\tau$ to approach the original function. See Figure 2 for an example of a soft relu surrogate. Note that softening a function may involve significantly altering the output domain, e. g., a soft surrogate for boolean operations outputs a probability instead of a Boolean value, likewise a soft surrogate for an argmax operation outputs a probability distribution over indices instead of an index.

**Straight-through estimation.** Unfortunately, blindly replacing a function with a soft surrogate may introduce undesired effects in the forward pass, e. g., resulting in unphysical rollout trajectories of a simulator. Fortunately, these problems can be addressed effectively through *straight-through estimation (STE)*.

Historically, STE dates back to early work on single-layer perceptrons (Rosenblatt, 1958), where the derivative of binary activations was replaced with the derivative of the

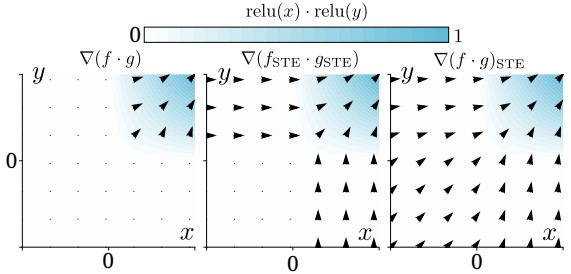

*Figure 3.* Arrows depict the normalized gradient of the product of two functions $f(x, y) = \text{relu}(x)$ and $g(x, y) = \text{relu}(y)$. Applying straight-through estimation on each function individually causes the gradient $\nabla(f_{\text{STE}} \cdot g_{\text{STE}})$ to be zero if $x < 0$, whereas the gradient $\nabla(f \cdot g)_{\text{STE}}$ is non-zero.

identity map. Following Bengio et al. (2013), STE keeps the original function in the forward pass during automatic differentiation, but uses the gradient of a soft surrogate in the backward pass. In our framework, STE is implemented via the straight-through trick, writing

$$f_{\text{STE}}(x) = \text{sg}(f(x)) + f_\tau(x) - \text{sg}(f_\tau(x)), \qquad (1)$$

where sg denotes the stop-gradient operator of an automatic differentiation library. Therefore, we obtain $f_{\text{STE}} = f$ on the forward pass, but $\nabla f_{\text{STE}} = \nabla f_\tau$ on the backward pass. In practice, using a soft surrogate for differentiation amounts to wrapping a soft function like `sj.relu` in the `sj.st` function decorator, which implements the straight-through trick, i.e., `y_soft_st = sj.st(sj.relu)`. Figure 2 illustrates straight-through wrapping a soft relu surrogate.

**The straight-through pitfall.** A subtle issue arises when STE-wrapped functions interact multiplicatively. Automatic differentiation computes the gradient between the product of functions $f_{\text{STE}}(x)$ and $g_{\text{STE}}(y)$ via the product rule as

$$\nabla(f_{\text{STE}} \cdot g_{\text{STE}}) = \nabla f_\tau \cdot g + f \cdot \nabla g_\tau, \qquad (2)$$

where the original functions $f$ and $g$ appear as multiplicative gates in the backward pass. This defeats the purpose of softening, as the gradient can still vanish when scaled by $f$ or $g$. As illustrated in Figure 3 for the case of $f(x, y) := \text{relu}(x)$ and $g(x, y) = \text{relu}(y)$, $\nabla(f_{\text{STE}} \cdot g_{\text{STE}})$ is zero when $x, y < 0$. To avoid gradient multiplication by zero, STE should be applied to the *composite function* such that

$$\nabla(f \cdot g)_{\text{STE}} = \nabla(f_\tau \cdot g_\tau) = \nabla f_\tau \cdot g_\tau + f_\tau \cdot \nabla g_\tau. \quad (3)$$

In code, this reads: (notice the decorator)

```python
@sj.st
def relu_prod(x, y, **kwargs):
    relu_x = sj.relu(x, **kwargs)
    relu_y = sj.relu(y, **kwargs)
    return relu_x * relu_y
```

To our knowledge, this "STE pitfall" and its remedy have not been explicitly discussed before.

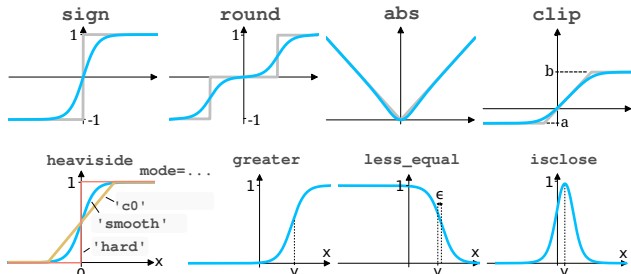

*Figure 4.* **Top:** The heaviside function surrogates $H_\tau$ are used to derive the sign, round, abs, and clip functions. **Bottom:** Comparison operations compare values $x, y \in \mathbb{R}$ by interpreting $H_\tau$ as a CDF defining the probability that $x > 0$.

## 3. Elementwise Operators

Conceptually, all our elementwise soft operators, some of which are illustrated in Figure 4, are rooted in the relaxation of the *Heaviside step function,* defined as

$$H(x) := \begin{cases} 0 & \text{if } x < 0, \\ 0.5 & \text{if } x = 0, \\ 1 & \text{else.} \end{cases} \qquad (4)$$

Its derivative is zero everywhere except at the origin, where it is not defined. Following many previous works (Funahashi, 1989; Han & Moraga, 1995; Rojas, 1996; Mhaskar, 1996; Duch & Jankowski, 1999; Huang et al., 2006b;a; Yuen et al., 2021; Li et al., 2021), we soften the Heaviside function with a sigmoidal function that traces a characteristic S-shape. Some examples include the standard exponential-based sigmoid (referred to as `smooth` mode)

$$H_\tau(x) := \sigma\left(\frac{x}{\tau}\right) = \frac{1}{1 + \exp(-x/\tau)}, \qquad (5)$$

or the piece-wise polynomial-based sigmoidals[1]

$$H_\tau(x) := \begin{cases} 0 & \text{if } x < -\tau, \\ g\left(\frac{x}{\tau}\right) & \text{if } -\tau \leq x \leq \tau, \\ 1 & \text{else,} \end{cases} \qquad (6)$$

with $g_{\text{c0}}(s) := \frac{1}{2} + \frac{s}{2}$, $g_{\text{c1}}(s) := \frac{1}{2} + \frac{3s}{4} - \frac{s^3}{4}$, and $g_{\text{c2}}(s) := \frac{1}{2} + \frac{15s}{16} - \frac{5s^3}{8} + \frac{3s^5}{16}$. The piecewise modes transition on $[-\tau, \tau]$; in the library, their input is rescaled by $1/5$ so that all modes share an effective transition width of approximately $10\tau$, matching the smooth mode ($\sigma(\pm 5) \approx 0.007/0.993$). These piece-wise Heaviside relaxations are either continuous (linear, also referred to as `c0` mode), differentiable (`c1`), or twice differentiable (`c2`).

### 3.1. Sign, abs, and round

A soft surrogate for the sign function is obtained by the Heaviside function and subtracting a constant. Moreover, a

---

[1]We overload the notation here.

soft surrogate for the abs function is obtained by gating the soft sign function with $x$, reading

$$\text{sign}_\tau(x) := 2 \cdot \text{H}_\tau(x) - 1, \tag{7}$$

$$\text{abs}_\tau(x) := \text{sign}_\tau(x) \cdot x. \tag{8}$$

Following Semenov (2025), the round function is implemented as the difference between shifted sigmoidals

$$\text{round}_\tau(x) := \sum_{k=\lfloor x \rfloor - K}^{\lfloor x \rfloor + K} k \left[ \text{H}_\tau\left(x - k + \tfrac{1}{2}\right) - \text{H}_\tau\left(x - k - \tfrac{1}{2}\right) \right],$$

where $K$ controls the number of neighbouring sigmoidals that are taken into account. Larger temperatures $\tau$ require a higher $K$, as each sigmoidal is softened over a larger region, thereby increasing its region of influence.

### 3.2. ReLU and clip

Due to its widespread use in ML, the rectified linear unit (ReLU) deserves extra attention. The function $\text{relu}(x) := \max\{0, x\}$ has zero gradient for $x < 0$, and at $x = 0$ it is only sub-differentiable. To get a soft surrogate, we can transform the sigmoidal via Integration

$$\text{relu}_\tau(x) := \int_0^x \text{H}_\tau(t) \, dt, \tag{9}$$

or via a gating mechanism

$$\text{relu}_\tau(x) := x \cdot \text{H}_\tau(x). \tag{10}$$

Therefore, for each of the Heaviside relaxations, we define two ReLU relaxations, some of which are well known. For instance, integrating the sigmoid yields the Softplus $\tau \log(1 + e^{x/\tau})$. On the other hand, gating the sigmoid recovers the SiLU activation (Hendrycks & Gimpel, 2016) $x \cdot \sigma(x/\tau)$. From the soft ReLU, we can now derive a soft surrogate for the clip function as

$$\text{clip}_\tau(x, a, b) := a + \text{relu}_\tau(x - a) - \text{relu}_\tau(x - b). \tag{11}$$

### 3.3. Comparison operators and differentiable logic

Next, we turn to soft surrogates of comparison operators, such as greater or equal. The hard operators output a Boolean variable $b \in \{\text{True}, \text{False}\}$, whereas our soft surrogates output a value in the interval $[0, 1]$. We directly get the soft comparison operators via the soft Heaviside function

$$\begin{aligned}
\text{greater}_\tau(x, y) &= \text{H}_\tau(x - y - \epsilon), \\
\text{gtr\_equal}_\tau(x, y) &= \text{H}_\tau(x - y + \epsilon), \\
\text{less}_\tau(x, y) &= \text{H}_\tau(y - x - \epsilon), \\
\text{less\_equal}_\tau(x, y) &= \text{H}_\tau(y - x + \epsilon), \\
\text{equal}_\tau(x, y) &= 2 \cdot \text{less\_equal}_\tau(\text{abs}_\tau(x - y), 0), \\
\text{isclose}_\tau(x, y) &= 2 \cdot \text{less\_equal}_\tau(\text{abs}_\tau(x - y), \text{tol}).
\end{aligned} \tag{12}$$

Here, $\epsilon$ denotes the machine precision to ensure that the surrogates converge to the correct hard function as $\tau \to 0^+$. In the definition of $\text{isclose}_\tau$, we use a standard tolerance $\text{tol} = \text{atol} + \text{rtol} \cdot \text{abs}_\tau(y)$ with $\text{atol} > 0$ and $\text{rtol} > 0$ denoting the relative and absolute tolerance.

The output of our soft surrogates can be interpreted as the probability of the hard comparison being True. For the sake of simplicity, we refer to such a probability as a soft Boolean or, in short, a "SoftBool". This perspective has been studied extensively in the field of fuzzy logic (Belohlavek et al., 2017). Given multiple SoftBools $p_1, \ldots, p_n$, differentiable logic operators reduce to the manipulation of probabilities. By default, our framework implements the logical all operator as

$$\text{all}(p_1, \ldots, p_n) := \prod_{j=1}^n p_j, \tag{13}$$

denoting the joint probability that independent Boolean events $p_1, \ldots, p_n$ equate to True. Alternatively, we allow using the geometric mean providing advantageous numerical scaling for gradient computation. Similarly, the not operator yields the probability that a Boolean event is False

$$\text{not}(p) := \neg p = 1 - p. \tag{14}$$

Other fuzzy logical operators are derived using the all and not operators

$$\begin{aligned}
\text{any}(p_1, \ldots, p_n) &:= \neg \, \text{all}(\neg p_1, \ldots, \neg p_n), \\
\text{and}(p, q) &:= \text{all}(p, q), \\
\text{or}(p, q) &:= \text{any}(p, q), \\
\text{xor}(p, q) &:= \text{or}\big(\text{and}(p, \neg q), \text{and}(\neg p, q)\big).
\end{aligned} \tag{15}$$

Similar to how normal Boolean variables can be used to select elements among two arrays, we can use a SoftBool variable $p_i$ to do a soft selection via the expectation

$$z_i = p_i \cdot x_i + (1 - p_i) \cdot y_i. \tag{16}$$

In code, this is abstracted from the user via

```
soft_cond = sj.greater(x, y) # [0.05 0.73 0.98]
z = sj.where(soft_cond, x, y) # [0.39 0.27 0.69]
```

## 4. Axiswise Operators

After discussing elementwise operators in the previous section, this section discusses and derives soft surrogates for axis-wise operators such as argmax, sort or argtopk.

The simplest example of an axiswise operation is the (arg)max operator. The standard argmax operator returns an index $j \in \{0, \ldots, n - 1\}$, and the max operator simply selects the corresponding element $x_j$. Another way to write

this is as an inner product $x_j = \mathbf{e}_j \cdot \mathbf{x}$, where $\mathbf{e}_j$ is one at index $j$ and zero everywhere else. We can now relax the notion of an index as an indicator vector to a "SoftIndex" on the unit simplex

$$\Delta_n := \big\{ \mathbf{p} \in [0,1]^n \mid \sum_{0 \le j < n} p_j = 1 \big\}. \tag{17}$$

In the case of the $\mathrm{argmax}$ operator, a simple relaxation that produces a SoftIndex is the softmax function (more accurately referred to as *softargmax*). The previous dot-product index-selection can now be interpreted as an expectation

$$\mathbf{p} \cdot \mathbf{x} = \sum_{0 \le j < n} p_j x_j = \mathop{\mathbb{E}}_{j \sim \mathbf{p}}[x_j]. \tag{18}$$

In code, our drop-in replacements read as:

```
x = jnp.array([0.1, 0.4, 0.8])

idx = jnp.argmax(x) # 2
y = jax.lax.dynamic_index_in_dim(x, idx) # [0.8]

soft_idx = sj.argmax(x) # [0.004 0.042 0.953]
y = sj.dynamic_index_in_dim(x, soft_idx) # [0.78]
```

We can further generalize this technique of using SoftIndices to sorting and ranking. To see this, we first note that sorting an array $\mathbf{x} \in \mathbb{R}^n$ can be viewed as multiplying it with the sorting permutation matrix

$$P^\star \in \Sigma = \{P \in \{0,1\}^{n \times n} \mid P\mathbf{1}_n = \mathbf{1}_n, P^\top \mathbf{1}_n = \mathbf{1}\}_n \tag{19}$$

such that

$$\mathrm{sort}(\mathbf{x}) = P^\star \mathbf{x}. \tag{20}$$

Throughout, $\mathrm{sort}$ is in ascending order. The rows of $P^\star$ are indicator vectors, and the matrix multiplication can be interpreted as doing index selection in parallel. In this sense, the permutation matrix $P^\star$ can be interpreted as the generalized $\mathrm{argsort}$ operator, i. e.,

$$\arg \mathrm{sort}(\mathbf{x}) = P^\star. \tag{21}$$

Similarly, the same matrix can be used for ranking, where rank 1 is assigned to the largest element,

$$\mathrm{rank}(\mathbf{x}) = (P^\star)^\top [n, \dots, 1]^\top. \tag{22}$$

We can now relax the notion of a permutation matrix into a bistochastic matrix, the set of which is also called the Birkhoff polytope

$$\mathbb{B}_n = \{P_\tau \in \mathbb{R}_+^{n \times n} \mid P_\tau \mathbf{1}_n = \mathbf{1}_n, P_\tau^\top \mathbf{1}_n = \mathbf{1}_n\}. \tag{23}$$

Note that in practice, for sorting or ranking, we only need $P_\tau$ to be a row- or column-stochastic matrix, respectively. The following two sections discuss how we can compute

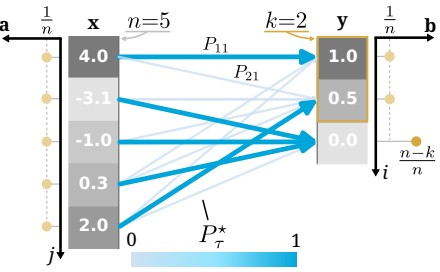

*Figure 5.* OT-based $\arg\mathrm{topk}$ computes a (scaled) optimal transport plan $P_\tau^\star$ whose entries $P_{ij}^\star$ contain the probability mass transported from the $j$-th entry of $\mathbf{x}$ to the $i$-th entry of the anchor $\mathbf{y}$.

soft permutation matrices $P_\tau$ from $\mathbf{x}$, either resorting to regularized optimal transport (OT) or unit simplex projections. Finally, we will discuss an approach that directly relaxes the sort operator via projection onto the permutahedron without relaxing permutation matrices. In this version of the library, we deliberately choose to focus on deterministic approaches, and do not consider stochastic ones such as (Berthet et al., 2020; Jang et al., 2017; Maddison et al., 2017; Paulus et al., 2020). Table 2 provides an overview on the various available methods and regularizations.

### 4.1. Optimal Transport-based

The first family of algorithms is based on optimal transport. In general, optimal transport describes optimally moving probability mass from one distribution to another distribution while paying minimal cost. As described by Cuturi et al. (2019), regularized OT can be used to define soft (arg)sort operators. The general idea is to mimic a sorting-like behavior by defining two uniform distribution, one on the to-be-sorted input vector $\mathbf{x} \in \mathbb{R}^n$ and one on a set of increasing "anchor" points $\mathbf{y} \in \mathbb{R}^n$. The optimal transport cost matrix is then defined as the squared distance between input and anchor points, which means that probability mass from large values of $\mathbf{x}$ will be moved to large anchor points. As the anchor points are sorted, this creates a sorting behavior, which is illustrated in Figure 5.

**OT problem.** The regularized optimal transport problem between two discrete probability measures $\sum_{i=1}^{n} a_i \delta_{x_i}$, with source vector $\mathbf{x} = [x_1, x_2, \dots, x_n]^\top$ and probability weight vector $\mathbf{a} = [a_1, a_2, \dots, a_n]^\top$, and $\sum_{j=1}^{m} b_j \delta_{y_j}$, with target vector $\mathbf{y} = [y_1, y_2, \dots, y_m]^\top$ and probability weight vector $\mathbf{b} = [b_1, b_2, \dots, b_m]^\top$, is defined as

$$\Gamma_\tau^\star := \arg \min_{\Gamma \in U(\mathbf{a}, \mathbf{b})} \langle \Gamma, C_{\mathbf{xy}} \rangle + \tau R(\Gamma). \tag{24}$$

Here, the cost matrix $(C_{\mathbf{xy}})_{ij} = (x_i - y_j)^2$ is computed as the squared differences and $U$ denotes the transport polytope of matrices summing to the marginals across rows and columns, i. e.,

$$U(\mathbf{a}, \mathbf{b}) := \{\Gamma \in \mathbb{R}_+^{n \times m} \mid \Gamma \mathbf{1}_m = \mathbf{a}, \Gamma^\top \mathbf{1}_n = \mathbf{b}\}. \tag{25}$$

Note that we omit the dependency of the optimal transport plan $\Gamma_\tau^\star$ on $(\mathbf{a}, \mathbf{x}, \mathbf{b}, \mathbf{y})$ for brevity.

We consider the case of an entropic regularizer $R(\Gamma) = \sum_{i,j} \Gamma_{ij}(\log \Gamma_{ij} - 1)$ (Cuturi, 2013), Euclidean regularizer $R(\Gamma) = \frac{1}{2} \sum_{i,j} \Gamma_{ij}^2$ (Blondel et al., 2018) and p-norm regularizer $R(\Gamma) = \frac{1}{p} \sum_{i,j} |\Gamma_{ij}|^p$ (Paty & Cuturi, 2020). It is well known that the Euclidean regularizer has a sparsity-inducing effect on the gradients; see, e. g., SparseMax (Martins & Astudillo, 2016). As discussed by Sander et al. (2023) in the context of projections onto the permutahedron, the p-norm regularizer shares this property, but additionally allows for everywhere differentiable relaxations (Theorem 1). Interestingly, for univariate inputs, the closed-form solutions of the entropic and Euclidean OT problem reduce to the corresponding sigmoidal functions used in our soft Heaviside relaxation, as shown in Section D.3.

**OT based operators.** We set the source marginal to a uniform distribution $\mathbf{a} = \mathbf{1_n}/n$, and specifically for sorting and ranking, we follow Cuturi et al. (2019) by setting uniform $\mathbf{b} = \mathbf{1_n}/n$ and grid-like $\mathbf{y} = \frac{1}{n-1}[0, \ldots, n-1]^\top$.

We define the scaled transposed transportation plan $P_\tau^\star := n(\Gamma_\tau^\star)^\top$. Since $\Gamma_\tau^\star \mathbf{1} = \mathbf{a}$ and $(\Gamma_\tau^\star)^\top \mathbf{1} = \mathbf{b}$, each row and column of $P_\tau^\star$ sums to 1. This allows us to interpret $P_\tau^\star$ as a soft permutation matrix.

Moreover, we first standardize and then squash $\mathbf{x}$ with a sigmoid to the range $[0, 1]$. Although this is not strictly necessary, the use of $\mathbf{x}$ and $\mathbf{y}$ in the range $[0, 1]$ improves numerical stability and makes the softening parameter independent of the scale of $\mathbf{x}$. Then we have[2]

$$\arg \operatorname{sort}_\tau(\mathbf{x}) = P_\tau^\star \in \mathbb{R}^{n \times n}, \qquad (26)$$

$$\operatorname{sort}_\tau(\mathbf{x}) = P_\tau^\star \mathbf{x} \in \mathbb{R}^n, \qquad (27)$$

$$\operatorname{rank}_\tau(\mathbf{x}) = (P_\tau^\star)^\top [n, \ldots, 1]^\top \in \mathbb{R}^n. \qquad (28)$$

To abstract the soft indices away from the user, we offer several replacements of utility functions for index selection that mirror the behavior of standard hard JAX and PyTorch index selection, e. g.,

```
x = jnp.array([0.3, 1.0, -0.5])

ind = jnp.argsort(x) # [2 0 1]
val = jnp.take_along_axis(x, ind) # [-0.5  0.3  1.0]

soft_ind = sj.argsort(x)
    # [[0.07 0.    0.93 ], [...], [...]]
val = sj.take_along_axis(x, soft_ind)
    # [-0.444  0.31   0.936]
```

In principle sorting is enough to define many other operators like max, median and top-k, via appropriate selection and

---

[2]Note, that the multiplication of the soft permutation matrix with $\mathbf{x}$ has close ties to the gating-style definition of the soft relu function in Equation (10), see Section D.5 for details.

combination of the soft-sorted values. However, the dimensionality of the cost matrix for an $n$-dimensional vector $\mathbf{x}$ is $n \times n$, which can lead to large memory requirements.

Fortunately, it is possible to strongly reduce the dimensionality, by transporting multiple elements of $\mathbf{x}$ which we are not interested in (e. g., the bottom $n - k$ elements in top-k) to a shared dummy anchor point, thereby reducing the number of anchors. For top-k, we follow Xie et al. (2020) by setting $\mathbf{b} = [\frac{1}{n}, \ldots, \frac{1}{n}, \frac{n-k}{n}]^\top \in \mathbb{R}^{1 \times (k+1)}, \mathbf{y} = \frac{1}{k}[k, \ldots, 1, 0]^\top \in \mathbb{R}^{1 \times (k+1)}$, then

$$\arg \operatorname{topk}_\tau(\mathbf{x}, k) = (P_\tau^\star)_{1:k} \in \mathbb{R}^{k \times n}, \qquad (29)$$

$$\operatorname{topk}_\tau(\mathbf{x}, k) = (P_\tau^\star)_{1:k}\mathbf{x} \in \mathbb{R}^k. \qquad (30)$$

Note that we can also trivially define an OT-based max as $\operatorname{topk}_\tau(\mathbf{x}, k = 1)$; and min by flipping signs. For q-quantiles we slightly vary the approach of Cuturi et al. (2019) by setting $k = \lfloor q(n-1) \rfloor$, $\mathbf{b} = [\frac{k}{n}, \frac{1}{n}, \frac{1}{n}, \frac{n-k-2}{n}]^\top, \mathbf{y} = \frac{1}{3}[0, 1, 2, 3]^\top$, then we can compute the lower $\downarrow$ and upper $\uparrow$ q-quantiles via the second and third entry of $P_\tau^\star$, i. e.,

$$\operatorname{quantile}_\tau^\downarrow(\mathbf{x}, q) = (P_\tau^\star)_2 \mathbf{x} \in \mathbb{R}, \qquad (31)$$

$$\operatorname{quantile}_\tau^\uparrow(\mathbf{x}, q) = (P_\tau^\star)_3 \mathbf{x} \in \mathbb{R}. \qquad (32)$$

Various quantile definitions can be recovered by appropriately combining the upper and lower quantiles (e. g., interpolating or using midpoint). We also trivially get the median by evaluating the quantile function with $q = 0.5$.

### 4.2. Approximate OT: unit simplex projection-based

The relaxations based on OT have many desirable properties such as smoothness in the case of an entropic regularizer. However, solving the OT problem can be memory- and compute-intensive in practice. We now describe different soft surrogates based on unit-simplex projections, which have a closed-form solution. The downside is that these relaxations typically still allow for some non-differentiabilities, e. g., when ties occur. However, in practice they often perform well and are therefore selected as the default method of choice.

**Unit simplex projection.** The regularized linear program over the simplex, see Equation (17), is defined as

$$\Pi_\tau(\mathbf{x}) := \arg \max_{\mathbf{p} \in \Delta_n} \langle \mathbf{x}, \mathbf{p} \rangle - \tau R(\mathbf{p}). \qquad (33)$$

We consider Euclidean regularization $R(\mathbf{p}) = \frac{1}{2}\|\mathbf{p}\|_2^2$, which leads to a Euclidean projection onto the unit simplex and entropic regularization $R(\mathbf{p}) = \sum_j p_j \log p_j$, leading to a closed-form solution via the softmax $\exp(\mathbf{x}/\tau)/\sum_j \exp(x_j/\tau)$. Finally, we consider p-norm regularization $R(\mathbf{p}) = \frac{1}{p} \sum_j |p_j|^p$, which leads to a Bregman projection (Bregman, 1967) onto the unit simplex.

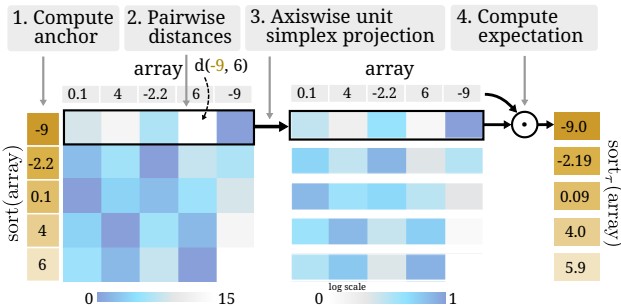

*Figure 6.* Illustration of the Softsort algorithm.

**SoftSort.** Entropy-regularized OT can be solved efficiently via the Sinkhorn algorithm (Cuturi, 2013). In the case of OT for sorting in Equation (26), where both marginals **a** and **b** are uniform, the Sinkhorn algorithm boils down to alternating row and column softmax-normalization of the cost matrix. SoftSort (Prillo & Eisenschlos, 2020) can be seen as approximating this algorithm by reducing it to a single row-wise softmax normalization via a clever initialization of the anchors. This corresponds to the entropic regularizer in Equation (33), which we generalize to other regularizers, leading to new variants of SoftSort.

To define the anchors, instead of using a uniform grid over the interval $[0, 1]$ as in Equation (26), the anchors are constructed by first applying the hard operator to $x$, e.g., $\mathbf{y} = \text{sort}(\mathbf{x})$. The cost matrix is the absolute difference between elements of $\mathbf{x}$ and $\mathbf{y}$, which will intuitively assign a low transportation cost between elements of $\text{sort}(\mathbf{x})$ that are close to $\mathbf{x}$. For the sort operator, this leads to

$$\arg \text{sort}_\tau(\mathbf{x}) := \Pi_\tau\Big(-|\text{sort}(\mathbf{x})\mathbf{1}_n^\top - \mathbf{1}_n\mathbf{x}^\top|\Big). \quad (34)$$

The projection is applied row-wise. The computation is illustrated in Figure 6. We showcase the output and partial derivatives of the soft sort operation for two different regularizers over a range of $\tau$ in Figure 7.

Note that the computation is independent over rows, we can therefore independently get soft operators for each of the sorted elements, which trivially gives us a soft (arg)topk/ quantile/max operator, by replacing the sort function in Equation (34) with the corresponding operator. Interestingly, the argmax operator reduces to $P_\tau(\mathbf{x})$, which for entropic regularization is simply softmax, thereby recovering the example used in the beginning of the chapter. For Euclidean regularization, it recovers the SparseMax (Martins & Astudillo, 2016) operator.

Finally, we can also define a novel SoftSort-style rank operator. For this we observe that $\Pi_\tau$ is applied independently to each row of the cost matrix, so the resulting matrix is only row-stochastic and the corresponding argsort operator cannot be transposed to define a soft ranking as we did in the OT case. Instead, we transpose the cost matrix before applying the unit-simplex projection, which gives distributions

over rank indices as

$$\arg \text{rank}_\tau(\mathbf{x}) := \Pi_\tau\Big(-|\mathbf{1}_n \text{sort}(\mathbf{x})^\top - \mathbf{x}\mathbf{1}_n^\top|\Big). \quad (35)$$

Note, that all SoftSort-based surrogates are not fully smooth, as the computation involves the hard sort operator (non-smooth at ties) and the absolute value function (non-smooth at zero). For this reason, the NeuralSort-based surrogates (which are fully smooth when combined with our soft abs) are selected as the default method for sort, rank, quantile, and median. However, for argmax, argmin, max, and min, the SoftSort-based method is the default, since it requires only a single simplex projection, giving $O(n)$ complexity compared to NeuralSort's $O(n^2)$, and in smooth mode it reduces to the standard softmax, the canonical soft argmax.

**NeuralSort.** NeuralSort (Grover et al., 2019) also utilizes unit-simplex projections but originates in a soft relaxation of the median operator. Defining the matrix of soft absolute differences $A_\mathbf{x}^\tau \in \mathbb{R}^{n \times n}$ using our soft abs from Equation (8) as

$$(A_\mathbf{x}^\tau)_{ij} := \text{abs}_\tau(x_i - x_j), \quad (36)$$

we have that $A_\mathbf{x}^\tau \mathbf{1}_n$ is the vector of sums of soft absolute differences between each element and all other elements.[3] It is well-known that the median minimizes this sum for $\tau \to 0^+$, therefore we have

$$\arg \text{median}(x) := \arg \min_{\mathbf{p} \in \Delta_n} \langle A_\mathbf{x}^0 \mathbf{1}_n, \mathbf{p} \rangle. \quad (37)$$

By adding a regularizer similar to SoftSort Equation (33), we get a soft surrogate

$$\arg \text{median}_\tau(x) := \arg \min_{\mathbf{p} \in \Delta_n} \langle A_\mathbf{x}^\tau \mathbf{1}_n, \mathbf{p} \rangle + \tau R(\mathbf{p}) \quad (38)$$

$$= \Pi_\tau(-A_\mathbf{x}^\tau \mathbf{1}_n). \quad (39)$$

NeuralSort generalizes this idea to the sorting operator, by setting

$$\arg \text{sort}_\tau(\mathbf{x})_i := \Pi_\tau\Big((2i - n - 1)\mathbf{x} - A_\mathbf{x}^\tau \mathbf{1}_n\Big). \quad (40)$$

For $\tau \to 0^+$ this recovers the true argsort operator. Intuitively, the vector that is projected is a linear combination of the original vector $\mathbf{x}$ and the vector of sums of soft absolute differences, which, when projected onto the unit simplex individually, give an argmax and an argmedian relaxation, respectively.

This also directly provides relaxations for the argmax, argmin, argquantile$^\uparrow$ and argquantile$^\downarrow$ operators by computing only the corresponding elements of the soft sorted

---

[3]The original NeuralSort paper (Grover et al., 2019) uses the hard $|x_i - x_j|$, which introduces gradient discontinuities at ties. By using our soft abs, we obtain fully smooth NeuralSort surrogates.

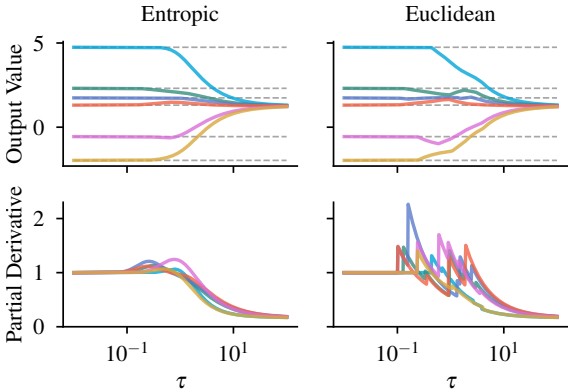

*Figure 7.* **Top:** Output of the sort operator using "softsort" for an array of $n = 6$ values as a function of softness $\tau$. Elements converge to the mean value as $\tau \to \infty$. **Bottom:** Partial derivatives $\partial y_j / \partial x_i$. Here, $j$ denotes the sorted position of input element $x_i$. As $\tau \to \infty$, the derivatives converge to $1/n$.

array, i.e., with $c^{\uparrow} = n - 1 - 2\lceil q(n-1) \rceil$ and $c^{\downarrow} = n - 1 - 2\lfloor q(n-1) \rfloor$,

$$\arg\max_{\tau}(x) \coloneqq \Pi_{\tau}\Big((n-1)\mathbf{x} - A_{\mathbf{x}}^{\tau}\mathbf{1}_n\Big) \quad (41)$$

$$\arg\min_{\tau}(x) \coloneqq \Pi_{\tau}\Big(-(n-1)\mathbf{x} - A_{\mathbf{x}}^{\tau}\mathbf{1}_n\Big) \quad (42)$$

$$\arg\text{quantile}_{\tau}^{\uparrow}(x, q) \coloneqq \Pi_{\tau}\big(-c^{\uparrow}\mathbf{x} - A_{\mathbf{x}}^{\tau}\mathbf{1}_n\big) \quad (43)$$

$$\arg\text{quantile}_{\tau}^{\downarrow}(x, q) \coloneqq \Pi_{\tau}\big(-c^{\downarrow}\mathbf{x} - A_{\mathbf{x}}^{\tau}\mathbf{1}_n\big) \quad (44)$$

Note that the original paper (Grover et al., 2019) only treats the smooth (entropic) projection; we extend it to c0, c1, and c2 regularizers. As with SoftSort, the NeuralSort argsort matrix is only row-stochastic. To obtain soft ranks, we normalize each column to sum to one and compute $\text{rank}_{\tau}(\mathbf{x}) = \widetilde{P}^{\top}[n, \dots, 1]^{\top}$, mirroring Equation (26).

### 4.3. Permutahedron projection-based

For sorting and ranking, both OT-based and unit-simplex-projection-based approaches involve computations with the $n \times n$ cost matrix. In SoftJAX, we avoid materializing this matrix by processing rows in chunks via lax.scan, reducing memory from $O(n^2)$ to $O(n)$ while also yielding $O(1)$ XLA compilation time. Nevertheless, for large $n$, the $O(n^2)$ time complexity remains a bottleneck. Blondel et al. (2020); Sander et al. (2023) resolve this issue by directly softening the value-space operators, e.g., sort instead of argsort. This is achieved by interpreting these operators as projections onto the permutahedron, the convex hull of permutations of $\mathbf{z}$, i.e.,

$$\mathcal{P}(\mathbf{z}) \coloneqq \text{conv}(\{\mathbf{z}_{\sigma} \mid \sigma \in \Sigma\}). \quad (45)$$

Here, $\Sigma$ is the set of permutation matrices, see Equation (19). The Euclidean projection of a vector $\mathbf{y} \in \mathbb{R}^n$ onto the

permutahedron of another vector $\mathbf{z} \in \mathbb{R}^n$ is given by

$$\text{Proj}_{\tau}(\mathbf{y}, \mathbf{z}) \coloneqq \arg\max_{\mathbf{p} \in \mathcal{P}(\mathbf{z})} \langle \mathbf{y}, \mathbf{p} \rangle - \frac{\tau}{2} \sum_j p_j^2 \quad (46)$$

$$= \arg\min_{\mathbf{p} \in \mathcal{P}(\mathbf{z})} \frac{1}{2} \Big\| \mathbf{p} - \frac{\mathbf{y}}{\tau} \Big\|_2^2. \quad (47)$$

Blondel et al. (2020) also propose a variant that uses an entropic regularizer by optimizing over exponentiated inputs and taking the log, i.e.,

$$\text{Proj}_{\tau}(\mathbf{y}, \mathbf{z}) \coloneqq \log\big(\arg\max_{\mathbf{p} \in \mathcal{P}(e^{\mathbf{z}})} \langle \mathbf{y}, \mathbf{p} \rangle - \tau \langle \mathbf{p}, \log \mathbf{p} - 1 \rangle\big).$$

**FastSoftSort.** Intuitively, FastSoftSort (Blondel et al., 2020) now projects the sorted ranks onto the permutahedron of the input $\mathbf{x}$, which will select the permutation of $\mathbf{x}$ that is closest to being sorted. In the same way, projecting the (negative) $\mathbf{x}$ onto the permutahedron of the possible ranks selects the ranks that follow the ordering of $\mathbf{x}$. This gives the surrogate operators

$$\text{sort}_{\tau}(\mathbf{x}) = \text{Proj}_{\tau}([1, \dots, n], \mathbf{x}), \quad (48)$$

$$\text{rank}_{\tau}(\mathbf{x}) = \text{Proj}_{\tau}(-\mathbf{x}, [1, \dots, n]). \quad (49)$$

Other operators like top-k, quantile, median or max can be computed by first running soft sorting, and then selecting the appropriate values.

Crucially, Blondel et al. (2020) derive an algorithm based on isotonic optimization which solves the projection onto the permutahedron in $O(n \log n)$. This is much faster than the runtime of Sinkhorn which is $O(T n^2)$, where $T$ is the number of Sinkhorn iterations. However, note that this approach does not allow for an argsort or argrank operator.

**SmoothSort.** In OT and unit simplex projection-based approaches, entropic regularization typically leads to dense Jacobians, whereas Euclidean regularization promotes sparsity, see e.g., (Blondel et al., 2018). However, in FastSoft-Sort this is not the case: While Euclidean regularization behaves as usual by promoting sparsity, the entropic regularization behaves very similarly to the Euclidean case by promoting sparsity, as also discussed by Blondel et al. (2020).

Therefore, we propose a new variant of this surrogate family by adding entropic regularization to a dual formulation of the permutahedron projection; see Section D.6 for details. This leads to a surrogate that behaves more like the entropic regularizer in the OT setting, which yields dense Jacobians. Moreover, by replacing the hard order-statistic bounds in the permutahedron LP with smooth log-sum-exp relaxations (Section D.6), this variant is $\mathcal{C}^{\infty}$ differentiable, unlike the other FastSoftSort variants which are at most $\mathcal{C}^2$. Unfortunately, we cannot use the same fast PAV algorithm in this

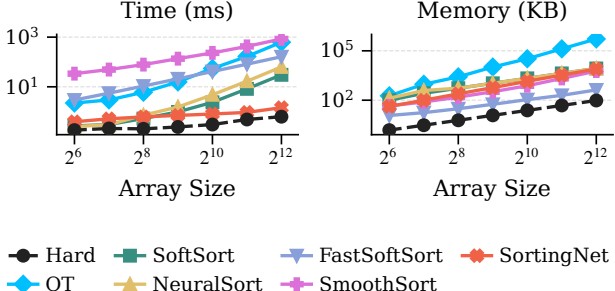

*Figure 8.* Computation time (**left**) and peak memory consumption (**right**) for a full forward-backward pass of `sj.sort` using `smooth` regularization. All six soft sorting methods are compared against the hard baseline. Extended results across all modes are in Figure 17.

case, because the smooth bounds require $O(n^2)$ preprocessing and the LP is solved via L-BFGS. However, compared to entropy-regularized OT, the $O(n^2)$ cost is paid only once (for the bounds), not per iteration, and we do not need to materialize an $n \times n$ matrix.

### 4.4. Sorting network-based

An alternative approach that avoids both the $O(n^2)$ pairwise cost matrix and iterative optimization is to use a differentiable sorting network. Following Petersen et al. (2021), who propose replacing the hard compare-and-swap operations in a bitonic sorting network with soft comparisons based on a logistic sigmoid, we implement a bitonic sorting network using our soft Heaviside function $H_\tau$ (Equations (5) and (6)) for soft swapping, i. e.,

$$\sigma = H_\tau(a - b), \qquad (50)$$
$$\text{soft\_min}(a, b) = \sigma \cdot b + (1 - \sigma) \cdot a, \qquad (51)$$
$$\text{soft\_max}(a, b) = \sigma \cdot a + (1 - \sigma) \cdot b. \qquad (52)$$

In `smooth` mode, $H_\tau$ is the logistic sigmoid, directly recovering the method of Petersen et al. (2021), while `c0`, `c1`, and `c2` modes yield new sorting network variants with the corresponding smoothness guarantees. Following Petersen et al. (2021), we obtain soft permutation matrices by tracking an $n \times n$ matrix $P$ (initialized to the identity) through the network. At each compare-and-swap step, the rows of $P$ corresponding to the two compared positions are mixed using the weights $\sigma$ and $1 - \sigma$, hence $\text{sort}(\mathbf{x}) = P\mathbf{x}$. Unlike Fast-SoftSort, sorting networks can therefore produce both sorted values and soft permutation matrices, enabling argsort and argmax in addition to sort, max, min, quantile, median, and top_k.

## 5. Benchmarks & Case Study

In order to be useful for machine learning practitioners, softened functions and their gradients must be both fast to compute and memory efficient. SoftJAX allows us to easily

compare these aspects across different methods within a unified framework. Figure 8 shows the computation time and peak memory of `sj.sort` for different input sizes and methods (`smooth` mode) on an Nvidia RTX 3060 GPU. The sorting network is the fastest soft method at $1.0\,\text{ms}$ for $n = 4096$ (only $\sim 3.8\times$ the hard baseline), followed by SoftSort ($16\,\text{ms}$) and NeuralSort ($37\,\text{ms}$). FastSoftSort is the most memory-efficient method, scaling roughly linearly ($420\,\text{kB}$ at $n = 4096$) since it avoids materializing a permutation matrix. SmoothSort and the OT-based surrogate are the slowest of the tested methods. Extensive benchmarking results on all methods and modes for a variety of axis-wise functions are included in Appendix E, and Appendix F distills these results into practical guidance for choosing among methods and modes.

Additionally, Appendix B collects case studies that demonstrate the ease of using SoftJAX in practice: softening a subroutine of MuJoCo XLA collision detection, four-digit MNIST sorting and quantile regression, a differentiable 3D mesh renderer, and minimal use-case examples covering top-$k$ feature selection and a differentiable rule-based classifier.

## 6. Conclusion

Standard tensor libraries with GPU acceleration and automatic differentiation have enabled differentiable optimization at scale in ML research. For many problems in ML and other fields, discrete primitives are needed, but result in uninformative gradients. Differentiable alternatives are scattered across papers, slowing progress and adoption. This work introduces SoftJAX and SoftTorch: feature-complete, unified and extensible libraries for soft relaxations of hard operators with principled straight-through estimation (STE).

We systematically derive differentiable surrogates for elementwise discrete operators from smooth relaxation of the Heaviside step function. Our framework also provides a broad set of axiswise differentiable discrete operators including argsort, argquantile, and argtop_k – implemented via state-of-the-art algorithms based on optimal transport and simplex/permutahedron projections and derive index-selection primitives (e.g., take_along_axis, choose) for end-to-end use. Through this standardization, this work derives a broad range of differentiable operators for which an implementation is currently not easily available (see Table 2).

Overall, the framework consolidates core components for *soft differentiable programming* into a single, well-tested, feature-complete library, improving reproducibility and lowering the barrier to widespread use of soft differentiable programming in practice.

## Acknowledgements

The authors thank Onur Beker for helpful discussions.

Additionally, we thank the International Max Planck Research School for Intelligent Systems (IMPRS-IS) for supporting Anselm Paulus.

The work on this paper was supported by the Czech Science Foundation (GAČR) grant no. 26-23981S.

This work was supported by the ERC - 101045454 REAL-RL and the German Federal Ministry of Education and Research (BMBF) through the Tübingen AI Center (FKZ: 01IS18039A). Georg Martius is a member of the Machine Learning Cluster of Excellence, EXC number 2064/1 – Project number 390727645.

## Impact Statement

By consolidating soft relaxation techniques into open-source libraries for JAX and PyTorch, this work lowers the barrier to applying gradient-based optimization in domains where hard, non-differentiable operations have traditionally required specialized expertise or ad-hoc workarounds. We expect the primary impact to be positive, as practitioners in robotics, scientific simulation, and structured prediction can more easily incorporate informative gradients into their pipelines. As with any tool that broadens access to optimization techniques, the libraries could in principle be applied to objectives whose optimization raises ethical concerns, but this risk is not specific to our contribution and is shared by all automatic differentiation frameworks.

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

## A. Library Overview

| Axiswise | Elementwise | Logical | Selection | Modes |
|---|---|---|---|---|
| argmin | heaviside | logical_and | where | ✓ smooth |
| min | round | logical_or | take_along_axis | ✓ c0 |
| argsort | sign | logical_xor | take | ✓ c1 |
| sort | abs | logical_not | choose | ✓ c2 |
| argquantile | relu | any | dynamic_index_in_dim | |
| quantile | clip | all | dynamic_slice_in_dim | **Methods** |
| argmedian | less | | dynamic_slice | ✓ Optimal Transport |
| median | less_equal | | | ✓ SoftSort |
| top_k | equal | | | ✓ NeuralSort |
| rank | not_equal | | | ✓ FastSoftSort |
| | isclose | | | ✓ SmoothSort |
| | | | | ✓ Sorting Network |

*Figure 9.* Overview of all operators implemented in SoftJAX. Redundant operators such as max or greater have been omitted for brevity. Additionally, SoftJAX provides autograd-safe wrappers for common mathematical functions whose standard implementations produce NaN gradients at boundary points: sqrt, arcsin, arccos, div, log, and norm. These wrappers clamp gradients near singularities, making them safe for use in differentiable pipelines. SoftTorch provides the same operators using PyTorch naming conventions; see Table 1 for the mapping.

*Table 1.* Naming differences between SoftJAX and SoftTorch. SoftTorch follows PyTorch conventions. Functions listed as "—" are not available in SoftTorch (no PyTorch equivalent).

| SoftJAX | SoftTorch |
|---|---|
| clip | clamp |
| equal | eq |
| top_k | topk |
| take_along_axis | take_along_dim |
| dynamic_index_in_dim | index_select |
| dynamic_slice_in_dim | narrow |
| choose | — |
| dynamic_slice | — |

*Table 2.* Overview of algorithms used. While not all regularization methods are detailed in the literature for the various algorithm types, we derived the missing combinations by systematically combining the above-listed works.

| | **Optimal transport** | | |
| --- | --- | --- | --- |
| | **smooth** | **c0** (Blondel et al., 2018) | **c1/c2** (Paty & Cuturi, 2020) |
| argmax | (Cuturi et al., 2019) | derived | derived |
| argsort | (Cuturi et al., 2019) | derived | derived |
| argquantile | (Cuturi et al., 2019) | derived | derived |
| argmedian | (Cuturi et al., 2019) | derived | derived |
| argtop_k | (Xie et al., 2020) | derived | derived |
| rank | (Cuturi et al., 2019) | derived | derived |
| | **SoftSort** | | |
| | **smooth** | **c0** | **c1/c2** |
| argmax | softmax | (Martins & Astudillo, 2016) | derived |
| argsort | (Prillo & Eisenschlos, 2020) | derived | derived |
| argquantile | (Prillo & Eisenschlos, 2020) | derived | derived |
| argmedian | (Prillo & Eisenschlos, 2020) | derived | derived |
| argtop_k | (Prillo & Eisenschlos, 2020) | derived | derived |
| rank | derived | derived | derived |
| | **NeuralSort** | | |
| | **smooth** | **c0** | **c1/c2** |
| argmax | (Grover et al., 2019) | derived | derived |
| argsort | (Grover et al., 2019) | derived | derived |
| argquantile | (Grover et al., 2019) | derived | derived |
| argmedian | (Grover et al., 2019) | derived | derived |
| argtop_k | (Grover et al., 2019) | derived | derived |
| rank | derived | derived | derived |
| | **FastSoftSort** | | |
| | **smooth** | **c0** | **c1/c2** |
| sort | (Blondel et al., 2020) | (Blondel et al., 2020) | (Sander et al., 2023) |
| rank | (Blondel et al., 2020) | (Blondel et al., 2020) | (Sander et al., 2023) |
| | **SmoothSort** | | |
| | **smooth** | **c0** | **c1/c2** |
| sort | novel | — | — |
| rank | novel | — | — |
| | **Sorting network** | | |
| | **smooth** | **c0** | **c1/c2** |
| sort | (Petersen et al., 2021) | derived | derived |
| argsort | (Petersen et al., 2021) | derived | derived |

## B. Case Studies

### B.1. Collision detection in MuJoCo XLA

Mujoco (Todorov et al., 2012) is one of the most widely used simulators throughout robotics research. MuJoCo XLA (MJX) is a re-implementation of MuJoCo using JAX to enable GPU-parallelized automatic differentiation of robot simulations. Unfortunately, neither MuJoCo nor MJX are softly differentiable as emphasized by Paulus et al. (2026). To improve differentiability, Paulus et al. (2026) introduced soft relaxations of MuJoCo's collision detection algorithms for simple primitives, i. e., sphere-plane and box-plane collisions. However, softening MuJoCo's collision detection for mesh-mesh collisions is significantly more challenging as it involves numerous nested combinations of non-differentiable discrete operations. Alternatively, Beker et al. (2025; 2026) propose efficient soft collision detection via signed distance fields. To illustrate the utility of SoftJAX, we soften one of the key subroutines in MJX collision detection. The subroutine as shown in Figure 12 (left) chooses four vertices (A, B, C, D) from an ordered polygon that roughly maximize the contact patch area.

To render this algorithm smoothly differentiable, we convert it to SoftJax as shown in Figure 12 (right). For this, the

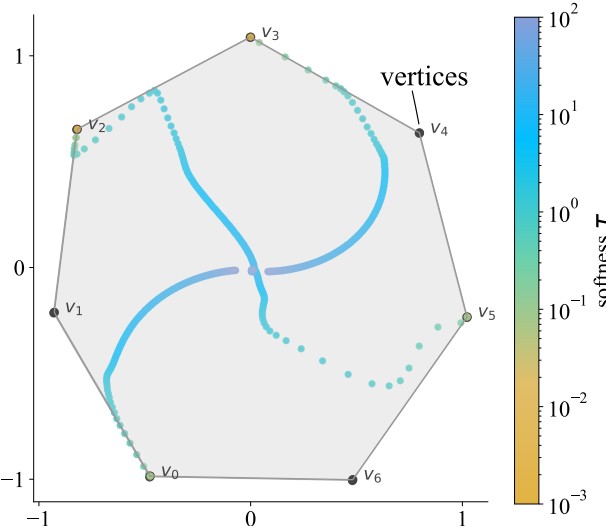

*Figure 10.* Manifold points computed by the algorithm's SoftJAX reimplementation over different softness values $\tau$. While a small softness causes the algorithm to choose points close to the vertices of the polygon, for large softness values, all four points collapse into a single point.

following changes are made:

1. Replace discrete `jnp.argmax` with `sj.argmax`, which returns a "SoftIndex" distribution,

2. Replace indexing with `sj.dynamic_index_in_dim`, which uses the "SoftIndex" as input,

3. Replace `jnp.abs` with `sj.abs`.

Note that the soft version returns "SoftIndex" distributions (shape: 4 x n) instead of the hard indices. The remaining code of the collision detection, therefore, needs to be adjusted accordingly.

**Effect of smoothing on point selection and gradients.** Figure 10 illustrates how the soft algorithm chooses four points from a seven-sided polygon as the softness parameter varies (computed as `soft_probs @ poly`). For small softness values, the soft selection closely matches the hard algorithm. As softness increases, the selected points meet in a single point as the `sj.argmax` reduces to the computation of a mean, as also shown in Figure 11 (Left).

A key motivation for softening is to obtain informative gradients with respect to all function inputs. As shown in Figure 11 (Right), MJX's hard version of manifold point selection always chooses four points out of the polygon's five vertices such that for one polygon point, the gradient is null. In comparison, the algorithm's soft re-implementation yields smooth, non-zero gradients at all vertices of the polygon (here deploying optimal transport with `smooth` regularization). Small softness values (e. g., $\tau = 0.01$) cause the algorithm to select points similarly to its hard counterpart at the expense of increased gradient magnitudes (note the logarithmic y-axis). When small softness values are required in an application, we recommend to use gradient clipping to improve numerical stability.

**Avoiding the STE pitfall.** We could now directly use one of the relaxations as a differentiable proxy. However, oftentimes it is desirable not to alter the forward pass. For instance, in a physics simulation, we do not want to alter the forward physics. In these cases, we can resort to the straight-through trick, which means to replace only the gradient of a hard function on the forward pass with the gradient of the soft function on the backward pass. To avoid the STE pitfall, the straight-through trick is applied on the outer level of the downstream function, which calls the whole function twice instead of each of the primitives:

```
from softjax.straight_through import st

manifold_points_softjax_st = st(manifold_points_softjax)
```

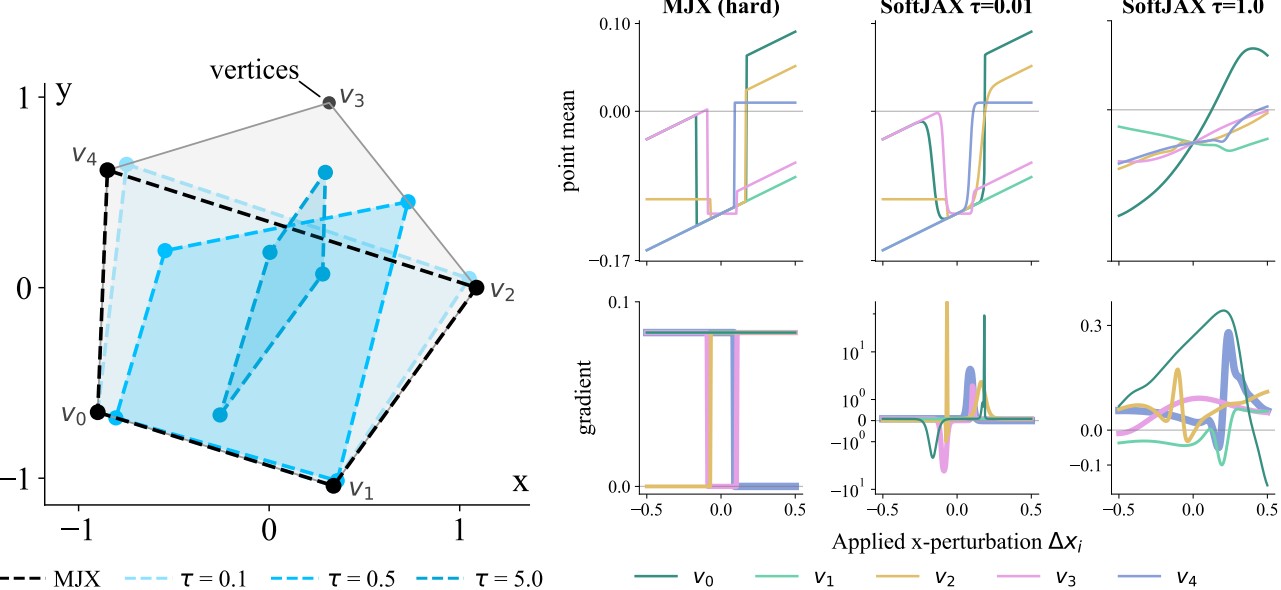

*Figure 11.* **Left:** The `manifold_points_mjx` algorithm selects the four vertices of a polygon (here $n = 5$) that maximize the enclosed area. In contrast, `manifold_points_softjax` selects the four points by computing an expectation using probability distributions over vertex indices. For small softness $\tau = 0.1$, SoftJAX selects points similar to MJX. **Right:** Perturbing the x-position of an individual vertex $v_i$ by $\Delta x_i$ changes the mean over all coordinates of the selected points. Notably, in the mode smooth with $\tau = 0.1$ and $\tau = 1$, the SoftJAX variant yields smooth gradients of the mean with respect to $\Delta x_i$. By contrast, MJX yields non-zero gradients solely for the vertices that have currently been selected by the algorithm.

```python
def manifold_points_mjx(poly, poly_mask, poly_norm:
        jnp.ndarray):

    dist_mask = jnp.where(poly_mask, 0.0, -1e6)
    # Note: We add a small tie-breaker
    # A: select the most penetrating vertex
    a_logits = dist_mask - 0.1 * jnp.arange(poly.shape
        [0])
    a_idx = jnp.argmax(a_logits)
    a = poly[a_idx]

    # B: farthest from A (largest squared distance)
    b_logits = ((a - poly) ** 2).sum(axis=1) +
        dist_mask
    b_idx = jnp.argmax(b_logits)
    b = poly[b_idx]

    # C: farthest from the AB line within the plane
    ab = jnp.cross(poly_norm, a - b)
    ap = a - poly
    c_logits = jnp.abs(ap.dot(ab)) + dist_mask

    c_idx = jnp.argmax(c_logits)
    c = poly[c_idx]

    # D: farthest from edges AC and BC
    ac = jnp.cross(poly_norm, a - c)
    bc = jnp.cross(poly_norm, b - c)
    bp = b - poly
    dist_bp = jnp.abs(bp.dot(bc)) + dist_mask

    dist_ap = jnp.abs(ap.dot(ac)) + dist_mask

    d_logits = dist_bp + dist_ap
    d_idx = jnp.argmax(d_logits)
    return jnp.array([a_idx, b_idx, c_idx, d_idx])
```

```python
def manifold_points_softjax(poly, poly_mask, poly_norm
        , softness: float = 0.1, mode: Literal["hard", "
        smooth"] = "smooth"):
    m, s = mode, softness
    dist_mask = jnp.where(poly_mask, 0.0, -1e6)

    # A: soft argmax over masked distances
    a_logits = dist_mask - 0.1 * jnp.arange(poly.shape
        [0])
    a_idx = sj.argmax(a_logits, mode=m, softness=s)
    a = sj.dynamic_index_in_dim(poly, a_idx, axis=0,
        keepdims=False)

    # B: soft argmax of distance from A
    b_logits = (((a - poly) ** 2).sum(axis=1)) +
        dist_mask
    b_idx = sj.argmax(b_logits, mode=m, softness=s)
    b = sj.dynamic_index_in_dim(poly, b_idx, axis=0,
        keepdims=False)

    # C: soft argmax farthest from AB line
    ab = jnp.cross(poly_norm, a - b)
    ap = a - poly
    c_logits = sj.abs(ap.dot(ab), mode=m, softness=s)
        + dist_mask
    c_idx = sj.argmax(c_logits, mode=m, softness=s)
    c = sj.dynamic_index_in_dim(poly, c_idx, axis=0,
        keepdims=False)

    # D: softargmax farthest from AC and BC
    ac = jnp.cross(poly_norm, a - c)
    bc = jnp.cross(poly_norm, b - c)
    bp = b - poly
    dist_bp = sj.abs(bp.dot(bc), mode=m, softness=s) +
        dist_mask
    dist_ap = sj.abs(ap.dot(ac), mode=m, softness=s) +
        dist_mask
    d_logits = dist_bp + dist_ap
    d_idx = sj.argmax(d_logits, mode=m, softness=s)
    return jnp.stack([a_idx, b_idx, c_idx, d_idx])
```

*Figure 12.* **Left:** Non-differentiable algorithm used by MJX for collision detection between mesh-mesh collisions. The algorithm chooses four vertices from a polygon that form a quadrilateral with approximately largest area. **Right:** Softened version of MJX's manifold point algorithm yielding informative gradients and returning four "SoftIndex" distributions.

## B.2. Sorting and quantile regression of four-digit MNIST

Sequences of four-digit MNIST numbers have been commonly used in literature to validate soft relaxations of sorting and quantile computation, see e. g., (Grover et al., 2019; Cuturi et al., 2019; Blondel et al., 2020; Petersen et al., 2021; 2022a). Following these works, each data sample contains a sequence of $n$ images depicting four-digit numbers ranging from 0000 to 9999, each of which, consists of four MNIST images (LeCun et al., 2010) stacked along the width dimension (yielding images of size $1 \times 28 \times 112$). For each four-digit image in the sequence, a CNN predicts a scalar, yielding the sequence of scalars $a_1, ..., a_n$. Depending on the task at hand the scalars are either fed to a differentiable argsort or quantile operator whose output is used to train the CNN.

The CNN architecture consists of two convolutional layers (32 and 64 filters of size $5 \times 5$ with padding 2, each followed by ReLU and $2 \times 2$ max-pooling), a fully connected layer with 64 hidden units and ReLU, and a linear output layer producing a single scalar. Training samples are generated on the fly by randomly drawing with replacement from a pool of 55,000 MNIST training digits to form the sequence of four-digit numbers; the remaining 5,000 digits form the validation pool and 10,000 test digits are used for evaluation. We train for 100,000 gradient steps using Adam with a learning rate of $0.001$ and a batch size of 100.

**Sorting.**    In this experiment, the sequence of scalars $a_1, ..., a_n$ is fed to SoftJAX's soft argsort operators. The argsort operation outputs a permutation matrix $P$ which is compared in a cross-entropy objective to the ground-truth permutation matrix $Q$ – defining the correct sorting operation for the four-digit sequence of numbers. Our reimplementation of this experiment is based on the code of (Petersen et al., 2022a) which has been ported to JAX using Equinox (Kidger & Garcia, 2021). Table 3 reports exact-match accuracy (proportion of sequences whose predicted ranking is entirely correct) and element-wise accuracy (proportion of individual positions ranked correctly).

**Quantile regression.**    Following (Prillo & Eisenschlos, 2020), instead of predicting the full ranking, the CNN outputs are fed to SoftJAX's quantile operator and is trained to predict the $q$-quantile (e.g., median for $q = 0.5$) of the scores. The soft quantile operator extracts a differentiable quantile value from the CNN scores, trained via mean squared error against the true quantile of the numerical labels. At evaluation time, the hard quantile is used. Table 4 reports MSE and in brackets the Spearman rank correlation between predicted and true quantile values across the test set.

*Table 3.* Results of training a CNN to sort four-digit MNIST sequences of length $n$. Reported metric is the percentage of correctly classified sequences while the brackets report the proportion of correctly identified individual element ranks.

| **smooth ($\tau = 0.1$)** | $n = 3$ | $n = 5$ | n=7 | n=9 | n=15 | n=32 |
|---|---|---|---|---|---|---|
| SoftSort | 93.0 (95.3) | 82.6 (92.2) | 69.5 (89.2) | 58.8 (87.3) | 24.0 (79.07) | 0.2 (62.1) |
| NeuralSort | 92.5 (94.9) | 81.7 (91.8) | 69.8 (89.4) | 58.6 (87.4) | 25.5 (79.7) | 0.0 (37.1) |
| Sorting Network | 92.6 (94.9) | 81.6 (91.8) | 67.5 (88.5) | 48.5 (84.0) | 16.6 (76.3) | – |
| OT | – | – | – | 49.9 (84.3) | 6.0 (66.1) | – |
| **c2 ($\tau = 0.1$)** | n=3 | n=5 | n=7 | n=9 | n=15 | n=32 |
| SoftSort | 88.0 (91.7) | 80.6 (91.5) | 66.4 (88.1) | 56.2 (86.8) | 30.2 (81.7) | |
| NeuralSort | 90.0 (93.2) | 78.2 (90.3) | 63.4 (87.1) | 49.2 (84.2) | 8.5 (69.7) | |
| Sorting Network | 90.3 (93.4) | 79.7 (90.9) | 65.2 (87.7) | 45.9 (82.9) | 13.1 (74.0) | |
| OT | 87.6 (91.5) | 84.1 (93.0) | 73.2 (90.7) | 58.0 (87.2) | 27.1 (80.7) | |
| **c0 ($\tau = 0.1$)** | n=3 | n=5 | n=7 | n=9 | n=15 | n=32 |
| SoftSort | 59.1 (70.9) | 16.4 (50.6) | 4.8 (44.2) | 0.8 (37.3) | – | – |
| Sorting Network | 74.9 (82.5) | 68.6 (85.8) | 49.1 (80.7) | 4.2 (51.3) | – | – |

*Table 4.* Results of training a CNN to predict the 0.5-quantile of four-digit MNIST sequences of length $n$. Reported metric is the MSE ($\times 10^{-4}$) when predicting the quantile, and in brackets, the Spearman rank correlation between predicted and true quantile values.

| **smooth ($\tau = 0.1$)** | n=3 | n=5 | n=7 | n=9 | n=15 |
|---|---|---|---|---|---|
| SoftSort | 30.9 (97.4) | 26.8 (96.9) | 21.4 (96.9) | 21.4 (96.4) | 15.7 (95.8) |
| NeuralSort | 26.8 (97.7) | 18.7 (97.8) | 24.2 (96.1) | 24.4 (95.4) | 16.3 (95.1) |
| Sorting Network | 31.9 (97.2) | 22.8 (97.6) | 25.1 (96.8) | 21.2 (96.8) | 26.4 (93.6) |

### B.3. Differentiable rendering

To illustrate the merits of SoftJAX and SoftTorch for quick prototyping, we implement a differentiable 3D mesh renderer in JAX. The implementation is based on Sebastian Lague's "Coding Adventure: Software Rasterizer" whose code is first ported to JAX and then extended to use soft operators drawing inspiration from Liu et al. (2019). This example is solely for the purpose of illustration, for professional differentiable rendering software, the interested reader is referred to Mitsuba 3.

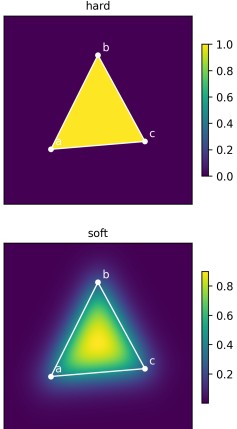

**Rasterization.** Given a triangle mesh $M$ and extrinsic variables (camera and lighting conditions), the vertices of each triangle are first projected onto the 2D image plane. For each projected triangle $f_j = \{a, b, c\}$, it is then determined for each 2D image pixel location $l_i$, if it lies inside the triangle. The probability $p_j^i$ indicating if $l_i$ lies inside $f_j$ is computed using Barycentric coordinates $b_a, b_b, b_c$ as

$$p_j^i = \text{all}\left(H_{\tau_1}(b_a), H_{\tau_1}(b_b), H_{\tau_1}(b_c)\right). \tag{53}$$

where $H_{\tau_1}$ denotes the soft Bool operator with softness $\tau_1$. The barycentric coordinate $b_a$ corresponds to the half signed area spanned by $c - b$ and $l_i - b$. In 2D, the signed area corresponds to the determinant of the matrix formed by $c - b$ and $l_i - b$ divided by two. In turn, given an image of size $H \times W$ and $N$ triangles, (53) gives rise to an array $P$ of size $N \times H \times W$ containing probabilities $p_j^i$. Further, the depth and color of all pixels relative to a given triangle are computed using clipped barycentric coordinates, giving rise to arrays of size $1 \times H \times W \times N$ and $3 \times H \times W \times N$.

*Figure 13.* $p_j^i$ of every pixel for "hard" and "soft" $H_{\tau_1}$.

**Z-buffering.** As next step, it is determined which of the triangles is close to the image pixels. To do so, following (Liu et al., 2019), the pixel-vertex-wise depth values $d_j^i$ are transformed to inverse depth $z_j^i$ such that 1 corresponds to the nearest vertex and 0 to the vertex that is furthest away from the respective pixel. In turn, a weight for each pixel-vertex pair is obtained through a weighted soft-argmax as

$$w_j^i = \frac{p_j^i \exp(z_j^i/\tau_2)}{\sum_k p_k^i \exp(z_k^i/\tau_2)}, \tag{54}$$

which we implemented using $\text{argmax}_{\tau_2}$ and noting that $b \exp(a) = \exp(a + \log(b))$ if $b > 0$. Note that the background was incorporated into (54) by appending an array of ones of size $1 \times H \times W$ to $P$ as well as appending an array of size $1 \times H \times W$ containing a small number $\epsilon$ to the arrays of inverse depths. Subsequently, given $w_j^i$, the color of each image pixel is obtained as $I^i = \sum_j w_j^i C_j^i$. Note that the color aggregation operation equals the computation of an expectation using the take_along_axis operator.

**Object pose estimation.** Figure 14 shows how our differentiable renderer optimizes object poses to minimize the L2 image loss between a "hard" target image and the "softly" rendered image estimate. The pose consists of the object's translational position and a nine dimensional rotation matrix representation based on the SVD decomposition as proposed by (Levinson et al., 2020) and discussed in detail in (Geist et al., 2024).

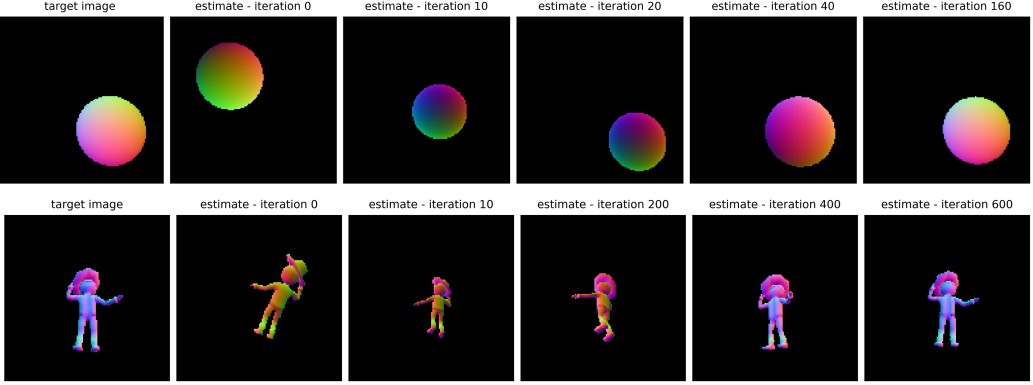

*Figure 14.* Gradient-based optimization of 3D object poses by minimizing the L2 errors between the hard target image and a softly rasterized image estimate. The objects are colored using a surface normal shader. Optimization uses AdamW ($\beta_1 = 0.9$, $\beta_2 = 0.99$) with stepsize 0.1 and softness parameters $\tau_1 = 0.1$, $\tau_2 = 0.1$. While the images show convergence for both objects, the loss landscape is highly non-linear and many initial positions did not converge. **Top:** Sphere object. **Bottom:** "Dave" figurine.

## B.4. Minimal use-case examples

In this section, we provide two minimal and self-contained examples that showcase how SoftJAX can be applied in practice.

**Top-k feature selection.**  A central challenge in applied machine learning is identifying the input features that most strongly influence model predictions (Guyon & Elisseeff, 2003). Given a trained model $f_\theta(x_1, \ldots, x_n)$, we seek a size-$k$ subset of features such that predictions are approximately preserved when all remaining features are masked. Formally, we seek the optimal index set

$$I^* = \underset{\substack{I \subseteq \{1,\ldots,n\} \\ |I|=k}}{\arg\min} \; \mathbb{E}_{x_1,\ldots,x_n \sim P} \left| f_\theta(x_1, \ldots, x_n) - f_\theta(\hat{x}_1, \ldots, \hat{x}_n) \right|, \tag{55}$$

where $P$ denotes the data distribution and

$$\hat{x}_i = \begin{cases} x_i & \text{if } i \in I \\ 0 & \text{otherwise} \end{cases} \tag{56}$$

denotes the masked feature vector.

Exact solution requires an exhaustive search over all $\binom{n}{k}$ subsets, which is computationally demanding. Instead, we assign a scalar *importance score* $g_i \in \mathbb{R}$ to each feature and optimize these scores via gradient descent. To this end, we use `sj.top_k` to obtain a differentiable soft mask over the top-$k$ scoring features. We evaluate on a linear model with sparse weights $w \in \mathbb{R}^{10}$ (only features 2, 4, 8 are non-zero). Gaussian noise is added to the targets to make the task more challenging and test robustness of the gradient signal. As shown in Figure 15(a), the smooth variant recovers the correct features in just a single step and preserves their ranking.

```
n_features, k = 10, 3
k1, k2 = jax.random.split(jax.random.PRNGKey(42))
X = jax.random.normal(k1, (100, n_features))
w_model = jnp.array([0, 2.0, 0, -1.5, 0, 0, 0, 5.0, 0, 0])
y = X @ w_model + 0.1 * jax.random.normal(k2, (100,))

def feature_selection_loss(g, X, y, w_model, mode="smooth"):
    _, soft_idx = sj.top_k(g, k=k, mode=mode, gated_grad=False)
    mask = soft_idx.sum(axis=0)
    y_pred = (X * mask) @ w_model
    return jnp.mean(sj.abs(y_pred - y))

g = jnp.zeros(n_features)
print("Hard grad:", jax.grad(feature_selection_loss)(g, X, y, w_model, mode="hard"))
print("Soft grad:", jax.grad(feature_selection_loss)(g, X, y, w_model, mode="smooth"))

for _ in range(5):
    g = g - 0.001 * jax.grad(feature_selection_loss)(g, X, y, w_model)
print("Selected features:", jax.lax.top_k(g, k=k)[1] + 1)
```

Output:

```
Hard grad: [0. 0. 0. 0. 0. 0. 0. 0. 0. 0.]
Soft grad: [2268.416 -2371.1378 2268.416 1126.1998 2268.416 2268.416 2268.416 -14633.9742 2268.416 2268.416]
Selected features: [8 2 4]
```

**Differentiable rule-based classifier.**  We demonstrate the utility of SoftJAX's logical and inequality operators by learning a simple rule-based classifier via gradient descent. Specifically, we aim to fit the model

$$f_{l,h}(\mathbf{x}) = (l < \mathbf{x}_1 < h) \lor (l < \mathbf{x}_2 < h), \tag{57}$$

where $l, h \in \mathbb{R}$ are learnable thresholds and $\mathbf{x} \in \mathbb{R}^2$ is a two-dimensional input with components $\mathbf{x}_1, \mathbf{x}_2$. This model corresponds to a fixed-structure decision tree whose internal splits are defined by $l$ and $h$. Classical approaches determine such thresholds by exhaustive or randomized search over candidate splits (cf. Breiman et al., 1984). SoftJAX, by contrast, renders the classifier differentiable, enabling $l$ and $h$ to be optimized directly via gradient descent. Figure 15(b, c) shows the thresholds quickly converging to a feasible solution within 10 steps, with accuracy reaching $100\%$.

We also demonstrate `@sj.st`: since decision trees typically produce hard class assignments, we decorate the loss with straight-through estimation, keeping the forward pass hard while routing gradients through the soft relaxation.

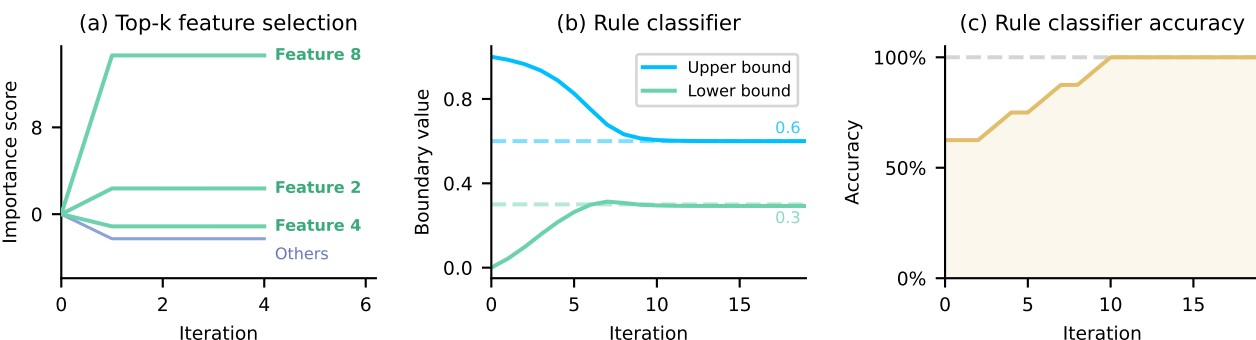

*Figure 15.* Optimization trajectories. **(a)** Learned importance scores for $n{=}10$ features; the three truly informative ones (Features 2, 4, 8) emerge after a single step. **(b, c)** Learned decision thresholds converging to a feasible solution and corresponding classification accuracy.

```python
x = jnp.array([[0.2, 0.8], [0.5, 0.3], [0.9, 0.1], [0.4, 0.7], [0.1, 0.4], [0.2, 0.7], [0.4, 0.1], [0.4, 0.7],
               [0.7, 0.29], [0.3, 0.3], [0.61, 0.25], [0.4, 0.6], [0.0, 0.1], [0.5, 0.3], [0.4, 0.9], [0.1,
                   0.57]])
labels = jnp.array([0.0, 1.0, 0.0, 1.0, 1.0, 0.0, 1.0, 1.0, 0.0, 1.0, 0.0, 1.0, 0.0, 1.0, 1.0, 1.0])

@sj.st
def rule_loss(params, x, labels, mode="smooth"):
    l, h = params[0], params[1]
    above = sj.greater(x, l, mode=mode)
    below = sj.less(x, h, mode=mode)
    in_range = sj.logical_and(above, below)
    preds = sj.any(in_range, axis=-1)
    return ((preds - labels) ** 2).sum()

params = jnp.array([0.0, 1.0])  # start with wide range [0, 1]
print("Hard grad:", jax.grad(rule_loss)(params, x, labels, mode="hard"))
print("Soft grad:", jax.grad(rule_loss)(params, x, labels, mode="smooth"))

for _ in range(20):
    params = params - 0.01 * jax.grad(rule_loss)(params, x, labels)
print("Learned [l, h]:", params)
```

Output:

```
Hard grad: [0. 0.]
Soft grad: [-4.2777  1.4152]
Learned [l, h]: [0.2925 0.5999]
```

## C. Input Standardization and Squashing

All axiswise operators in SoftJAX and SoftTorch optionally preprocess the input $\mathbf{x} \in \mathbb{R}^n$ via a *standardize-and-squash* transform, following Cuturi et al. (2019). This serves two purposes: (i) it maps $\mathbf{x}$ into $[0, 1]$, matching the range of the grid-like target marginal $\mathbf{y} = \frac{1}{n-1}[0, \ldots, n-1]^\top$ used in optimal transport, improving numerical stability; and (ii) it makes the softness parameter $\tau$ independent of the scale of $\mathbf{x}$.

**Forward transform.** Given the input $\mathbf{x}$, we first standardize and then squash with the logistic sigmoid $\sigma$, i.e.,

$$\mu = \text{mean}(\mathbf{x}), \quad s = \text{std}(\mathbf{x}), \quad \widetilde{\mathbf{x}} = \sigma\left(\frac{\mathbf{x} - \mu}{s}\right), \tag{58}$$

where $\mu$ and $s$ are computed along the specified axis. The transformed input $\widetilde{\mathbf{x}} \in (0, 1)^n$ is then passed to the axiswise operator.

**Inverse transform.** For value-returning operators (e.g., sort), the output $\widetilde{\mathbf{y}} \in (0, 1)^n$ is then mapped back to the original scale. For this we apply the inverse, by first unsquashing via the logit function, and then destandardizing, i.e.,

$$\mathbf{y} = \text{logit}(\widetilde{\mathbf{y}}) \cdot s + \mu, \quad \text{where} \quad \text{logit}(p) = \log \tfrac{p}{1-p}. \tag{59}$$

The exception is SmoothSort (Section 4.3), which skips standardize-and-squash. This is because it is the only method which can return values larger than the largest input element (or smaller than the smalles input element) as the output is not a convex combination of inputs. Hence with standardize-and-squash, the output would not always be in the interval $(0, 1)$, which can lead to undefined outputs of the unsquashing operation.

## D. Extended Axiswise Operators

### D.1. Solving optimization problems

**Optimal transport.** In SoftJAX, for `smooth`-mode OT, we use Sinkhorn iterations implemented via ott-jax (Cuturi et al., 2022) by default. Alternatively, L-BFGS as suggested in (Blondel et al., 2018), implemented via the Optimistix library (Rader et al., 2024), can be used. For `c0`, `c1`, and `c2` modes, we solve the dual OT problems (see Section D.2 for their formulations) using L-BFGS via Optimistix. Both libraries support implicit differentiation, which reduces memory usage and enables adaptive stopping criteria for optimization loops. In SoftTorch, we solve the optimal transport problems using POT (Flamary et al., 2021; 2024).

**Projection onto the unit simplex.** In `smooth` mode, this reduces to a simple call to a softmax function. In `c0` mode, we use the well-known $O(n \log n)$ algorithm of (Wang & Carreira-Perpiñán, 2013). For `c1` and `c2` modes, we derive closed-form solutions to the Bregman projection onto the unit simplex (Bregman, 1967), namely a quadratic formula for $p = 3/2$ (`c1`) and Cardano's method for $p = 4/3$ (`c2`), both running in $O(n \log n)$.

**Projection onto the permutahedron.** We follow (Blondel et al., 2020), who reduce the Euclidean and log-KL projections onto the permutahedron to isotonic optimization, which is in turn solved in $O(n \log n)$ via the pool adjacent violators (PAV) algorithm.

For SoftJAX, we reimplement the PAV algorithm in pure JAX, as the original NumPy code is incompatible with `jax.vmap`/`jax.jit`. In SoftTorch, we rely Numba-JIT for acceleration.

Moreover, we extend FastSoftSort to $p$-norm regularization following Sander et al. (2023). Note, that Sander et al. (2023) solve the PAV subproblem via bisection, we replace this with closed-form solutions to a quadratic formula for `c1` and Cardano's method for `c2`.

For SmoothSort (Section 4.3) we solve the dual of the entropically-regularized LP via L-BFGS (Rader et al., 2024). A custom VJP on the solver computes gradients via an adjoint saddle-point system. Note that SmoothSort is currently SoftJAX only.

## D.2. Dual OT problems

For marginals $\mathbf{a} \in \Delta_n$, $\mathbf{b} \in \Delta_m$, cost matrix $C \in \mathbb{R}^{n \times m}$, and dual potentials $\mathbf{f} \in \mathbb{R}^n$, $\mathbf{g} \in \mathbb{R}^m$, the dual of the entropy-regularized OT problem can be written as

$$\max_{\mathbf{f} \in \mathbb{R}^n, \mathbf{g} \in \mathbb{R}^m} \langle \mathbf{f}, \mathbf{a} \rangle + \langle \mathbf{g}, \mathbf{b} \rangle - \tau \sum_{ij} \exp\left(\frac{f_i + g_j - C_{ij}}{\tau}\right). \tag{60}$$

and the corresponding optimal primal solution can be recovered via

$$(\Gamma_\tau^\star)_{ij} = \exp\left(\frac{f_i^\star + g_j^\star - C_{ij}}{\tau}\right). \tag{61}$$

The dual of the Euclidean regularized OT problem can be written as

$$\max_{\mathbf{f} \in \mathbb{R}^n, \mathbf{g} \in \mathbb{R}^m} \langle \mathbf{f}, \mathbf{a} \rangle + \langle \mathbf{g}, \mathbf{b} \rangle - \frac{1}{2\tau} \sum_{ij} \left(f_i + g_j - C_{ij}\right)_+^2 \tag{62}$$

and the corresponding optimal primal solution can be recovered via

$$(\Gamma_\tau^\star)_{ij} = \frac{1}{\tau}\left(f_i^\star + g_j^\star - C_{ij}\right)_+. \tag{63}$$

The dual of the p-norm regularized OT problem can be written as

$$\max_{\mathbf{f} \in \mathbb{R}^n, \mathbf{g} \in \mathbb{R}^m} \langle \mathbf{f}, \mathbf{a} \rangle + \langle \mathbf{g}, \mathbf{b} \rangle - \frac{\tau^{1-q}}{q} \sum_{ij} \left(f_i + g_j - C_{ij}\right)_+^q, \tag{64}$$

where $q = \frac{p}{p-1}$. The corresponding optimal primal solution can be recovered via

$$(\Gamma_\tau^\star)_{ij} = \left(\frac{f_i^\star + g_j^\star - C_{ij}}{\tau}\right)_+^{\frac{1}{p-1}}. \tag{65}$$

## D.3. Elementwise operators as special case

While elementwise and axiswise soft operators are treated as distinct categories in the main text, they are closely related. Specifically, the elementwise soft surrogates described in Section 3 can be recovered as special cases of the axiswise operators when $n = 2$. Here, we demonstrate that applying axiswise operators to a two-element vector $\mathbf{x} = [0, x]^\top$ yields the soft Heaviside functions derived earlier. To distinguish the mode-specific sigmoidals in this section, we introduce superscripts for the smoothness class. The `smooth` mode is $\mathrm{H}_\tau^\infty(x) := \sigma(x/\tau) = e^{x/\tau}/(1 + e^{x/\tau})$. All piecewise modes share the structure

$$\mathrm{H}_\tau^k(x) := \begin{cases} 0 & \text{if } x < -\tau, \\ g^k(x/\tau) & \text{if } |x| \le \tau, \\ 1 & \text{if } x > \tau, \end{cases} \tag{66}$$

with interpolation functions $g^k : [-1, 1] \to [0, 1]$

$$g^0(s) = \tfrac{1}{2} + \tfrac{s}{2}, \qquad \text{(c0, piecewise linear, } \mathcal{C}^0) \tag{67}$$

$$g^1(s) = \tfrac{1}{2} + \tfrac{3s}{4} - \tfrac{s^3}{4}, \qquad \text{(c1, Hermite cubic, } \mathcal{C}^1) \tag{68}$$

$$g^2(s) = \tfrac{1}{2} + \tfrac{15s}{16} - \tfrac{5s^3}{8} + \tfrac{3s^5}{16}. \qquad \text{(c2, Hermite quintic, } \mathcal{C}^2) \tag{69}$$

The p-norm simplex projections at $n=2$ yield distinct $\mathcal{C}^1$ and $\mathcal{C}^2$ sigmoidals, which we denote $\widetilde{\mathrm{H}}_\tau^1$, $\widetilde{\mathrm{H}}_\tau^2$ (same piecewise structure with $\widetilde{g}^k$ replacing $g^k$)

$$\widetilde{g}^1(s) = \frac{\left(s + \sqrt{2 - s^2}\right)^2}{4}, \qquad (p=3/2 \text{ p-norm, } \mathcal{C}^1) \tag{70}$$

$$\widetilde{g}^2(s) = (t + s/2)^3, \quad t = -|s| \sinh\left(\tfrac{1}{3} \operatorname{arcsinh}\left(\tfrac{-2}{s^2|s|}\right)\right). \qquad (p=4/3 \text{ p-norm, } \mathcal{C}^2) \tag{71}$$

These smoothness classes are confirmed for general $n$ in Theorem 1.

**Reduction of soft argmax.** Consider the soft $\arg\max$ operator applied to $\mathbf{x} = [0, x]^\top$. The output is a probability distribution $\mathbf{p} \in \Delta_2$ over the indices. Since $p_0 + p_1 = 1$, the distribution is fully characterized by $p_1$, which is a sigmoidal function of $x/\tau$ for each regularizer, i. e.,

$$\arg\max{}_\tau([0, x]^\top) = [1 - p_1,\, p_1]^\top. \tag{72}$$

For entropic regularization, $\arg\max$ reduces to the standard softmax, and hence we recover the sigmoid

$$p_1 = \sigma(x/\tau) = \mathrm{H}_\tau^\infty(x). \tag{73}$$

For $p$-norm regularization, the projection of $[0, x]^\top$ onto the unit simplex solves $\arg\min_{\mathbf{p}\in\Delta_2} -\langle [0,x]^\top, \mathbf{p}\rangle + \frac{\tau}{p}\sum_i p_i^p$.

With $p = 2$ (c0 mode), the first-order condition gives

$$p_1 = \tfrac{1}{2} + \tfrac{x}{2\tau} = \mathrm{H}_\tau^0(x), \tag{74}$$

directly recovering the elementwise c0 sigmoidal.

With $p = 3/2$ (c1 mode), we get the first-order condition $\sqrt{p_1} - \sqrt{1-p_1} = x/\tau$, which can be solved via the quadratic formula by $p_1 = \widetilde{\mathrm{H}}_\tau^1(x)$. This is another sigmoidal function in $\mathcal{C}^1$ with $p_1' = 0$ at $x = \pm\tau$.

With $p = 4/3$ (c2 mode), we get the first-order condition $p_1^{1/3} - (1-p_1)^{1/3} = x/\tau$, which reduces to a depressed cubic. That can be solved by Cardano's formula, giving $p_1 = \widetilde{\mathrm{H}}_\tau^2(x)$. This is yet another sigmoidal function in $\mathcal{C}^2$. Note that $\mathrm{H}^1$, $\mathrm{H}^2$ are not special cases of an axiswise operator, but provide the same smoothness guarantees with simpler expressions. Theorem 1 confirms that the $\mathcal{C}^1$ and $\mathcal{C}^2$ smoothness classes of $\widetilde{\mathrm{H}}_\tau^1$ and $\widetilde{\mathrm{H}}_\tau^2$ extend to the general $n$-dimensional simplex and transport polytope projections.

**Reduction of soft sort and OT.** Similarly, we can analyze the behavior of soft sorting and optimal transport for $n = 2$. Let the input be $\mathbf{x} = [0, x]^\top$ with sorted anchors $\mathbf{y} = [0, 1]^\top$ and balanced marginals $\mathbf{a} = \mathbf{b} = [\frac{1}{2}, \frac{1}{2}]^\top$. The cost matrix is denoted as $C_{ij} = (x_i - y_j)^2$. The OT solution then takes the form

$$P_\tau^\star = \begin{pmatrix} p_1 & 1 - p_1 \\ 1 - p_1 & p_1 \end{pmatrix}, \qquad \Gamma = P/2, \tag{75}$$

which only leaves one degree of freedom $p_1 \in [0, 1]$. The cost simplifies to $\langle C, \Gamma \rangle = \frac{1+x^2}{2} - x p_1$, hence the regularized OT problem reduces to a scalar optimization in $p_1$.

For entropy-regularized OT, the first-order condition $\log(p_1/(1-p_1)) = x/\tau$ gives

$$p_1 = \sigma(x/\tau) = \mathrm{H}_\tau^\infty(x), \tag{76}$$

recovering the logistic sigmoid, identical to the simplex projection in the previous paragraph. For Euclidean regularization ($p{=}2$), the first-order condition yields

$$p_1 = \tfrac{1}{2} + x/\tau = \mathrm{H}_{\tau/2}^0(x), \tag{77}$$

matching the simplex result $\mathrm{H}_\tau^0(x)$ up to a factor of 2 in softness. For $p{=}3/2$, the first-order condition gives

$$\sqrt{p_1} - \sqrt{1 - p_1} = \sqrt{2}\, x/\tau, \tag{78}$$

and for $p{=}4/3$,

$$p_1^{1/3} - (1 - p_1)^{1/3} = 2^{1/3}\, x/\tau, \tag{79}$$

recovering the results of the previous paragraph up to softness scaling as $\widetilde{\mathrm{H}}_{\tau/\sqrt{2}}^1(x)$ and $\widetilde{\mathrm{H}}_{\tau/\sqrt[3]{2}}^2(x)$, respectively.

**Softness convention.** In our sigmoidal definitions, the piecewise polynomial modes (Equation (6)) transition from zero to one on the interval $[-\tau, \tau]$. In contrast, the smooth mode reaches near-saturation only at $\approx \pm 5\tau$. Therefore, we introduce a scaling factor for the piecewise polynomial modes as $g(x/(5\tau))$, which widens their transition to $[-5\tau, 5\tau]$, matching that of the smooth mode.

For axiswise operators, we do not add any mode-dependent scaling constants. As a result, the same $\tau$ value produces different effective softness across modes and problem sizes $n$ for axiswise operators. For users needing comparable behavior across modes or sizes, we recommend measuring the normalized entropy $H(\mathbf{P})/\log n$ of the soft permutation matrix and calibrating $\tau$ accordingly.

### D.4. Smoothness of $p$-norm regularized projections

The preceding section verifies the smoothness classes $\mathcal{C}^1$ and $\mathcal{C}^2$ of the $p$-norm sigmoidals $\widetilde{\mathrm{H}}_\tau^1$ and $\widetilde{\mathrm{H}}_\tau^2$ for $n = 2$. We now prove that these smoothness classes hold for the general $n$-dimensional simplex projection and for the $p$-norm regularized optimal transport plan.

**Theorem 1** (Smoothness of $p$-norm regularized projections)**.** *Let $n \geq 2$, $1 < p \leq 2$, $\tau > 0$, and let $q = p/(p-1)$ be the conjugate exponent with $q$ a positive integer. Define $k := q - 2$.*

*(i)* Unit simplex. *The $p$-norm regularized projection $\Pi_\tau \colon \mathbb{R}^n \to \Delta_n$,*

$$\Pi_\tau(\mathbf{x}) := \arg\max_{\mathbf{p} \in \Delta_n} \langle \mathbf{x}, \mathbf{p} \rangle - \tfrac{\tau}{p} \|\mathbf{p}\|_p^p, \tag{80}$$

*is a $\mathcal{C}^k$ map. For entropic regularization $\sum_j p_j \log p_j$, the projection $\Pi_\tau$ is $\mathcal{C}^\infty$.*

*(ii)* Transport polytope. *For $m \geq 2$, marginals $\mathbf{a} \in \Delta_n$, $\mathbf{b} \in \Delta_m$ and cost matrix $C \in \mathbb{R}^{n \times m}$, the $p$-norm regularized optimal transport plan*

$$\Gamma_\tau^\star(C) := \arg\min_{\Gamma \in U(\mathbf{a}, \mathbf{b})} \langle \Gamma, C \rangle + \tfrac{\tau}{p} \sum_{ij} |\Gamma_{ij}|^p \tag{81}$$

*is a $\mathcal{C}^k$ map of $C$ at every cost matrix for which the support $\{(i,j) : \Gamma_{ij}^\star > 0\}$ forms a connected bipartite graph. For entropic regularization $\sum_{ij} \Gamma_{ij}(\log \Gamma_{ij} - 1)$, the transport plan $\Gamma_\tau^\star$ is a $\mathcal{C}^\infty$ map of $C$ unconditionally, since the support is always the full bipartite graph.*

*Moreover, the bound $\mathcal{C}^k$ is sharp: $\Pi_\tau$ is not $\mathcal{C}^{k+1}$, and $\Gamma_\tau^\star$ is not $\mathcal{C}^{k+1}$. In particular, `c0` ($p=2$) gives $\mathcal{C}^0$, `c1` ($p=3/2$) gives $\mathcal{C}^1$, `c2` ($p=4/3$) gives $\mathcal{C}^2$, and `smooth` gives $\mathcal{C}^\infty$.*

Since the cost matrix in the sorting application is $C_{ij} = (x_i - y_j)^2$, which is $\mathcal{C}^\infty$ in $\mathbf{x}$, the transport plan $\Gamma_\tau^\star$ inherits the same $\mathcal{C}^k$ regularity as a function of the input $\mathbf{x}$ by the chain rule. The connectivity condition is generically satisfied, i. e., it holds for all cost matrices outside a set of measure zero.

*Proof.* The proof is based on the implicit function theorem applied to the optimality conditions of the optimization problems. Both parts rely on the positive-part power function, defined as $\varphi \colon \mathbb{R} \to [0, \infty)$ by $\varphi(t) := (t/\tau)_+^{k+1}$, of which all derivatives up to order $k$ exist and are continuous, therefore $\varphi \in \mathcal{C}^k(\mathbb{R})$.

**Part (i): Unit simplex.** The KKT conditions for the simplex-constrained problem are

$$x_i - \tau(p_i^\star)^{p-1} - \nu + \mu_i = 0, \quad 1 \leq i \leq n, \tag{82}$$

$$\sum_{i=1}^n p_i^\star = 1, \tag{83}$$

$$p_i^\star \geq 0, \quad \mu_i \geq 0, \quad \mu_i\, p_i^\star = 0, \tag{84}$$

where $\nu$ is the multiplier for the equality constraint $\mathbf{1}^\top \mathbf{p} = 1$ and $\mu_i$ are the multipliers for the non-negativity constraints. When $p_i^\star > 0$, complementary slackness gives $\mu_i = 0$, so the stationarity condition reduces to $p_i^\star = ((x_i - \nu)/\tau)^{1/(p-1)} =$

$\varphi(x_i - \nu)$. When $p_i^\star = 0$, the stationarity condition gives $\mu_i = \nu - x_i \geq 0$, i.e., $x_i \leq \nu$. In both cases $p_i^\star = \varphi(x_i - \nu)$, and substituting into the equality constraint yields

$$\sum_{i=1}^{n} \varphi(x_i - \nu) = 1. \tag{85}$$

Define $h \colon \mathbb{R} \times \mathbb{R}^n \to \mathbb{R}$ by $h(\nu, \mathbf{x}) := \sum_{i=1}^{n} \varphi(x_i - \nu) - 1$. As a finite sum of translates of $\varphi$, the function $h$ is $\mathcal{C}^k$ in $(\nu, \mathbf{x})$. The partial derivative with respect to $\nu$ is

$$\frac{\partial h}{\partial \nu} = -\sum_{i=1}^{n} \varphi'(x_i - \nu), \tag{86}$$

which is strictly negative at any $(\nu, \mathbf{x})$ satisfying $h = 0$. Indeed, $\sum_i \varphi(x_i - \nu) = 1 > 0$ implies that at least one index $i$ has $x_i > \nu$, and for any such index $\varphi'(x_i - \nu) > 0$.

For $k \geq 1$ (i.e., $p < 2$), the Implicit Function Theorem for $\mathcal{C}^k$ maps yields that $\nu$ is a $\mathcal{C}^k$ function of $\mathbf{x}$. Furthermore, the implicit $\nu(\mathbf{x})$ is unique and globally defined, since $h(\cdot, \mathbf{x})$ is strictly decreasing (it satisfies $\partial h / \partial \nu < 0$ at every root). Each component $p_i^\star(\mathbf{x}) = \varphi(x_i - \nu(\mathbf{x}))$ is then $\mathcal{C}^k$ as a composition of $\mathcal{C}^k$ functions, establishing $\Pi_\tau \in \mathcal{C}^k$. For entropic regularization, $\Pi_\tau(\mathbf{x}) = \exp(\mathbf{x}/\tau)/\sum_j \exp(x_j/\tau)$, which is $\mathcal{C}^\infty$.

**Part (ii): Transport polytope.** The argument follows the same structure as Part (i). From the dual formulation (Section D.2), the optimal transport plan satisfies $\Gamma_{ij}^\star = \varphi(f_i^\star + g_j^\star - C_{ij})$, where $(\mathbf{f}^\star, \mathbf{g}^\star)$ are the dual potentials. The marginal constraints $\sum_j \varphi(f_i + g_j - C_{ij}) = a_i$ and $\sum_i \varphi(f_i + g_j - C_{ij}) = b_j$ define, after fixing $g_m = 0$ to break the shift symmetry, a $\mathcal{C}^k$ implicit system of $n + m - 1$ equations in $n + m - 1$ unknowns. The Jacobian is non-singular if and only if the bipartite support graph $\{(i, j) : \Gamma_{ij}^\star > 0\}$ is connected, which is the standard non-degeneracy condition for optimal transport. Under this condition, the implicit function theorem gives $\Gamma_\tau^\star(C) \in \mathcal{C}^k$. For entropic regularization, $\Gamma_{ij}^\star = \exp((f_i^\star + g_j^\star - C_{ij})/\tau) > 0$ for all $(i, j)$, so the support is always connected and the same argument yields $\Gamma_\tau^\star \in \mathcal{C}^\infty$.

**Sharpness.** For any $n \geq 2$, restricting $\Pi_\tau$ to inputs of the form $\mathbf{x} = (x_1, x_2, -M, \ldots, -M)$ with $M$ large enough that only the first two components are active reduces the projection to the $n = 2$ case. The preceding section shows that this two-dimensional projection is the sigmoidal $p_1 = \widetilde{\mathrm{H}}_\tau^k(x)$, whose $(k+1)$-th derivative is discontinuous at $|x| = \tau$. Since this is a restriction of $\Pi_\tau$, the full map cannot be $\mathcal{C}^{k+1}$. The same embedding applies to $\Gamma_\tau^\star$ via the $n = m = 2$ reduction shown in the preceding section. $\qquad\square$

### D.5. Gating- vs integration-like behavior of axiswise operators

In the main text, we have seen that there are two principled ways to get a soft `relu` surrogate from a soft sigmoidal, either by integration

$$\mathrm{relu}_\tau(x) := \int_0^x \mathrm{H}_\tau(t) \, \mathrm{d}t, \tag{87}$$

which, in the case of the `smooth` regularizer, is simply a Softplus function, or via a gating mechanism

$$\mathrm{relu}_\tau(x) := \mathrm{H}_\tau(x) \cdot x, \tag{88}$$

which, in the case of an `smooth` regularizer, is simply a SiLU function,

We now observe that a similar mechanism is at play for axiswise operators: To get a soft surrogate for the `sort` operator, we "gate" the `argsort` operator with the input, i.e.

$$\mathrm{sort}_\tau(\mathbf{x}) := \arg\mathrm{sort}_\tau(\mathbf{x})\mathbf{x} = P_\tau^\star(\mathbf{x})\,\mathbf{x}. \tag{89}$$

Intuitively, this means that the `sort` operator defined in this way behaves qualitatively similar to the SiLU function, which has a characteristic minimum, in contrast to the monotonically increasing Softplus function.

A natural question is whether a version of the sort operator exists that behaves similarly to the Softplus function. Unfortunately, this would require integrating the multivariate argsort function. However, we are often anyway mainly interested in the *gradient* of the soft sort surrogate, while leaving the forward pass hard via straight-through estimation. The gradient (or, more precisely, the Jacobian) is extremely simple to compute. It is the function we would integrate, i. e., the argsort operator matrix.

In code, similar to the straight-through trick, we can manipulate our sort implementation to return the Jacobian of the integration-style version by using a stop-gradient operation on the argsort operator, i. e.,

```
soft_idx = jax.lax.stop_gradient(sj.argsort(x))
sorted_values = sj.take_along_axis(x, soft_idx)
```

This option is available via a simple flag, and it can be easily combined with straight-through estimation to get the hard forward pass and integration-style backward pass, e. g.,

```
sorted_values_st = sj.sort_st(x, gated_grad=False)
```

### D.6. Entropy-regularized dual of permutahedron projection

**Primal LP.** The optimization problem over the permutahedron as stated in the main text is given by

$$\max_{\mathbf{p} \in \mathcal{P}(\mathbf{z})} \langle \mathbf{y}, \mathbf{p} \rangle. \tag{90}$$

We can equivalently characterize the permutahedron constraint $\mathbf{p} \in \mathcal{P}(\mathbf{z})$ through a set of majorization inequalities

$$\sum_{i=1}^{k} p_i^{\downarrow} \leq \sum_{i=1}^{k} z_i^{\downarrow} \quad \forall k \colon 1 \leq k < n \tag{91}$$

and the sum equality

$$\sum_{i=1}^{n} p_i = \sum_{i=1}^{n} z_i, \tag{92}$$

where $\mathbf{p}^{\downarrow}$ and $\mathbf{z}^{\downarrow}$ are sorted in descending order.

**Reduction to ordered cone.** W.l.o.g., we can restrict the search space to ordered $\mathbf{p}$ by enforcing

$$p_i \geq p_{i+1} \quad \forall i \colon 1 \leq i < n \tag{93}$$

and then reorder the optimal solution back via the inverse permutation that sorts $\mathbf{z}$. Crucially, on the ordered cone, the majorization inequalities become linear in $\mathbf{p}$

$$\sum_{i=1}^{k} p_i \leq \sum_{i=1}^{k} z_i^{\downarrow} \quad \forall k \colon 1 \leq k < n. \tag{94}$$

We can therefore rewrite the permutahedron LP as

$$\begin{aligned}
\max_{\mathbf{p} \in \mathbb{R}^n} & \langle \mathbf{y}^{\downarrow}, \mathbf{p} \rangle \\
\text{subject to } & A\mathbf{p} \leq \mathbf{b} \\
& \mathbf{1}^{\top} \mathbf{p} = c \\
& D\mathbf{p} \geq 0,
\end{aligned} \tag{95}$$

where we used $A \in \mathbb{R}^{(n-1) \times n}$, $\mathbf{b} \in \mathbb{R}^{(n-1)}$, $c \in \mathbb{R}$, $D \in \mathbb{R}^{(n-1) \times n}$, defined as

$$(A\mathbf{p})_k := \sum_{i=1}^{k} p_i \quad \forall 1 \le k < n \tag{96}$$

$$b_k := \sum_{i=1}^{k} z_i^{\downarrow} \quad \forall 1 \le k < n \tag{97}$$

$$c := \sum_{i=1}^{n} z_i \tag{98}$$

$$(D\mathbf{p})_i := p_i - p_{i+1} \quad \forall 1 \le i < n. \tag{99}$$

**Smoothing.** The LP above is non-smooth in two ways. First, computing the majorization bounds $b_k = \sum_{i=1}^{k} z_i^{\downarrow}$ requires sorting $\mathbf{z}$, which is not a smooth operation. Second, the LP itself is a linear program whose solution jumps between vertices. We address both by (i) replacing $\mathbf{b}$ with a smooth relaxation $\widetilde{\mathbf{b}}$, and (ii) adding entropic regularization on the inequality slack variables.

For the bounds, we can rewrite $b_k$ as an optimization problem over subsets of indices, i. e.,

$$b_k = \sum_{i=1}^{k} z_i^{\downarrow} = \max_{\substack{S \subseteq [n] \\ |S|=k}} \sum_{i \in S} z_i. \tag{100}$$

We can then replace the hard maximum with a log-sum-exp, which leads to a smooth relaxation, i. e.,

$$\widetilde{b}_k := \tau \log \sum_{\substack{S \subseteq [n] \\ |S|=k}} \exp\left(\frac{1}{\tau} \sum_{i \in S} z_i\right) = \tau \log e_k\left(e^{\mathbf{z}/\tau}\right), \tag{101}$$

where $e_k(\mathbf{x}) = \sum_{|S|=k} \prod_{i \in S} x_i$ sums all products of $k$ distinct elements from $\mathbf{x}$. In the limit $\tau \to 0^+$, we have $\widetilde{b}_k \to b_k$. Since $e_n(\mathbf{x}) = \prod_i x_i$, we directly get $\widetilde{b}_n = \sum_i z_i = c$ thereby preserving the equality constraint. Note also that the relaxed bounds satisfy $\widetilde{b}_k \ge b_k$, so the feasible set is a slight superset of the true permutahedron.

We next explain how the smoothed bounds can be computed in practice. For this we use the recurrence

$$E_0^{(j)} = 1, \quad E_k^{(0)} = 0 \text{ for } k \ge 1, \tag{102}$$

$$E_k^{(j)} = E_k^{(j-1)} + e^{z_j/\tau} \cdot E_{k-1}^{(j-1)}, \tag{103}$$

where $E_k^{(j)} = e_k(e^{z_1/\tau}, \ldots, e^{z_j/\tau})$. This requires $O(n^2)$ time runtime. Note that we work in log-space (logaddexp) for numerical stability and use optimal online checkpointing (Stumm & Walther, 2010) to reduce memory from $O(n^2)$ to $O(n\sqrt{n})$.

In order to smoothen the LP, we introduce slack variables and add entropic regularization onto them using $\phi(t) = t \log(t) - t$, i. e.,

$$\max_{\mathbf{p} \in \mathbb{R}^n, \mathbf{s} \in \mathbb{R}^{n-1}, \mathbf{d} \in \mathbb{R}^{n-1}} \langle \mathbf{y}^{\downarrow}, \mathbf{p} \rangle - \tau \sum_{k=1}^{n-1} \phi(s_k) - \tau \sum_{i=1}^{n-1} \phi(d_i)$$

$$\text{subject to } A\mathbf{p} + \mathbf{s} = \widetilde{\mathbf{b}} \tag{104}$$

$$\mathbf{1}^{\top}\mathbf{p} = c$$

$$D\mathbf{p} = \mathbf{d}.$$

**Dual problem and solution recovery.** The smoothed primal problem has the dual problem

$$\min_{\alpha \in \mathbb{R}^{n-1}, \beta \in \mathbb{R}^{n-1}, \nu \in \mathbb{R}} \langle \alpha, \widetilde{\mathbf{b}} \rangle + \nu c + \tau \sum_{k=1}^{n-1} e^{-\alpha_k/\tau} + \tau \sum_{i=1}^{n-1} e^{-\beta_i/\tau}$$

$$\text{subject to } \mathbf{y}^{\downarrow} - A^{\top}\alpha + D^{\top}\beta - \nu\mathbf{1} = 0. \tag{105}$$

We now proceed by first eliminating $\nu$ via the equality constraint

$$\mathbf{0} = (\mathbf{y}^\downarrow + D^\top \beta - \nu \mathbf{1})_n \Rightarrow \nu(\beta) = y_n^\downarrow + (D^\top \beta)_n. \tag{106}$$

Now we define

$$\mathbf{u}(\beta) = \mathbf{y}^\downarrow + D^\top \beta - \nu(\beta)\mathbf{1}, \tag{107}$$

which satisfies $u_n(\beta) = 0$. Note that have $\mathbf{u}(\beta) = A^\top \alpha$, which means

$$\alpha_k(\beta) = u_k(\beta) - u_{k+1}(\beta) \quad \forall k \colon 1 \le k < n. \tag{108}$$

Finally, substituting $(\alpha_k(\beta), \nu(\beta))$ into the dual gives an unconstrained optimization problem over $\beta$, which we can solve via L-BFGS.

In tha last step we recover the primal solution from the dual solution $(\alpha^\star, \beta^\star, \nu^\star)$. This is done via

$$\mathbf{s}^\star = \exp(-\alpha^\star/\tau) \tag{109}$$
$$\mathbf{d}^\star = \exp(-\beta^\star/\tau) \tag{110}$$

and $\mathbf{p}^\star$ is recovered by first defining prefix sums

$$r_k := \widetilde{b}_k - s_k^\star = \sum_{i=1}^{k} p_i^\star \quad \forall k \colon 1 \le k < n, \tag{111}$$

and then computing $\mathbf{p}^\star$ by differencing

$$p_1^\star = r_1 \tag{112}$$
$$p_k^\star = r_k - r_{k-1} \quad \forall k \colon 2 \le k < n \tag{113}$$
$$p_n^\star = c - r_{n-1}. \tag{114}$$

**Backward pass.** The solver uses a custom VJP that solves an adjoint system via conjugate gradients, with $O(n)$ cost per iteration due to the banded structure of the LP constraints. Gradients through the smooth bounds use standard reverse-mode autodiff through Equation (101).

## E. Additional Benchmark Experiments

We include additional benchmarking experiments for elementwise operators (Figure 16) and axiswise operators (Figures 17 to 23). Each figure shows runtime, JIT compilation time, and peak memory consumption as a function of input array size, broken down by regularization mode (`smooth`, `c0`, `c1`, `c2`).

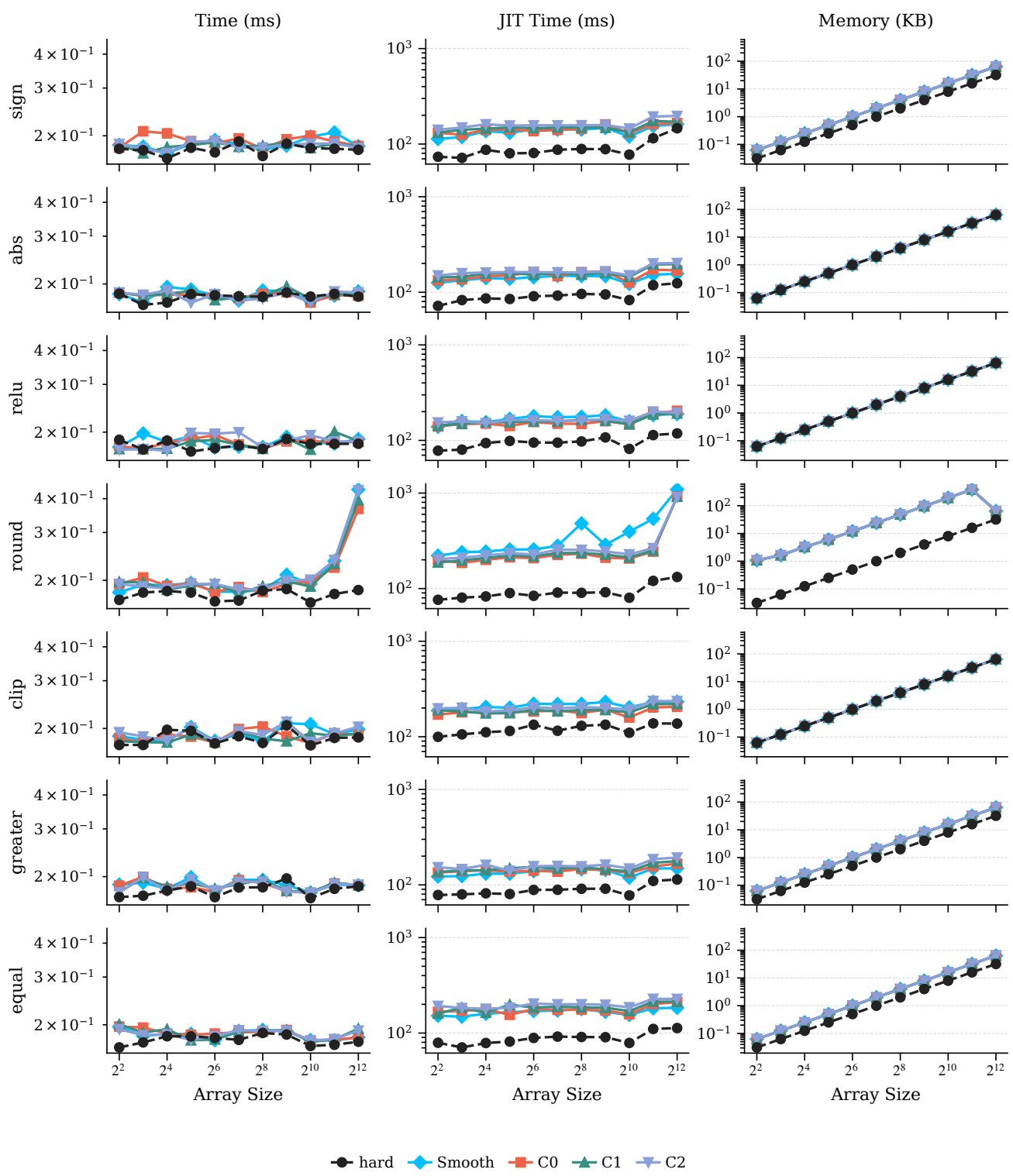

*Figure 16.* Benchmark results for elementwise operators.

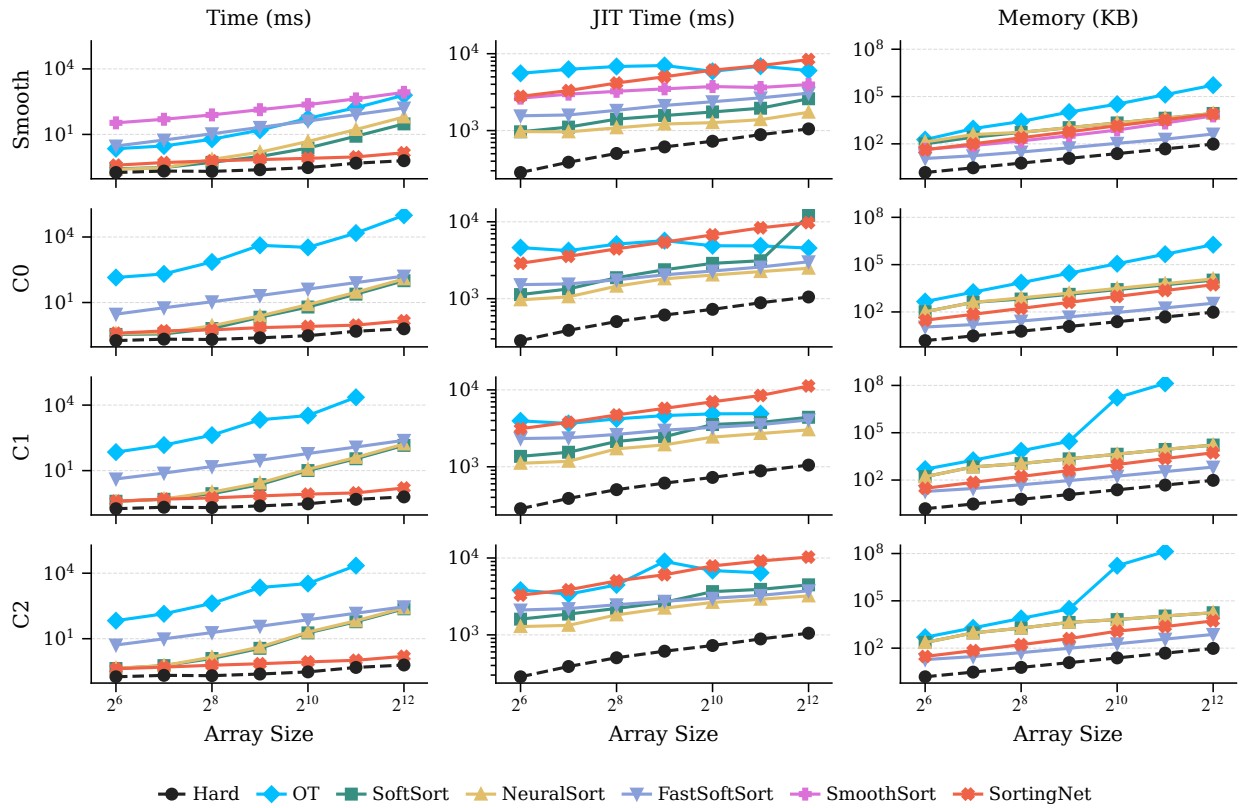

*Figure 17.* Benchmark results for `sj.sort`.

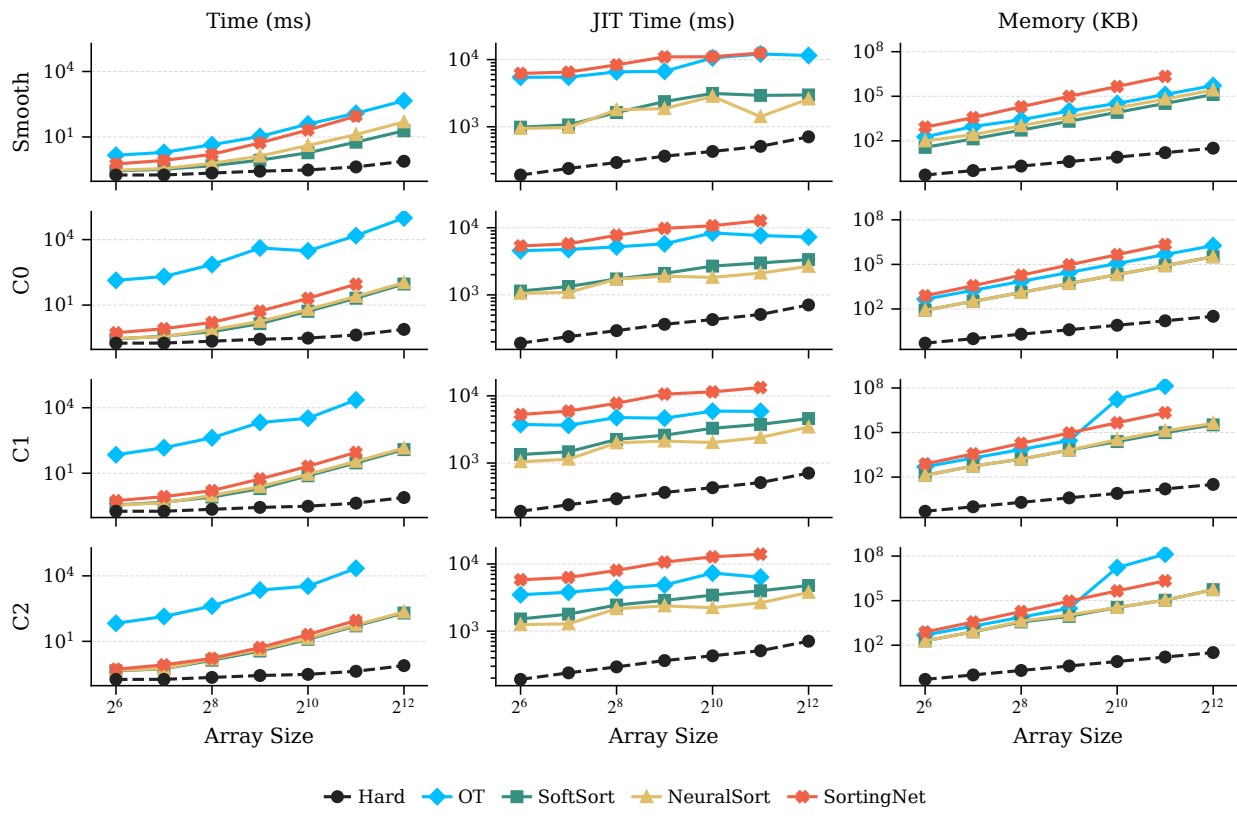

*Figure 18.* Benchmark results for `sj.argsort`.

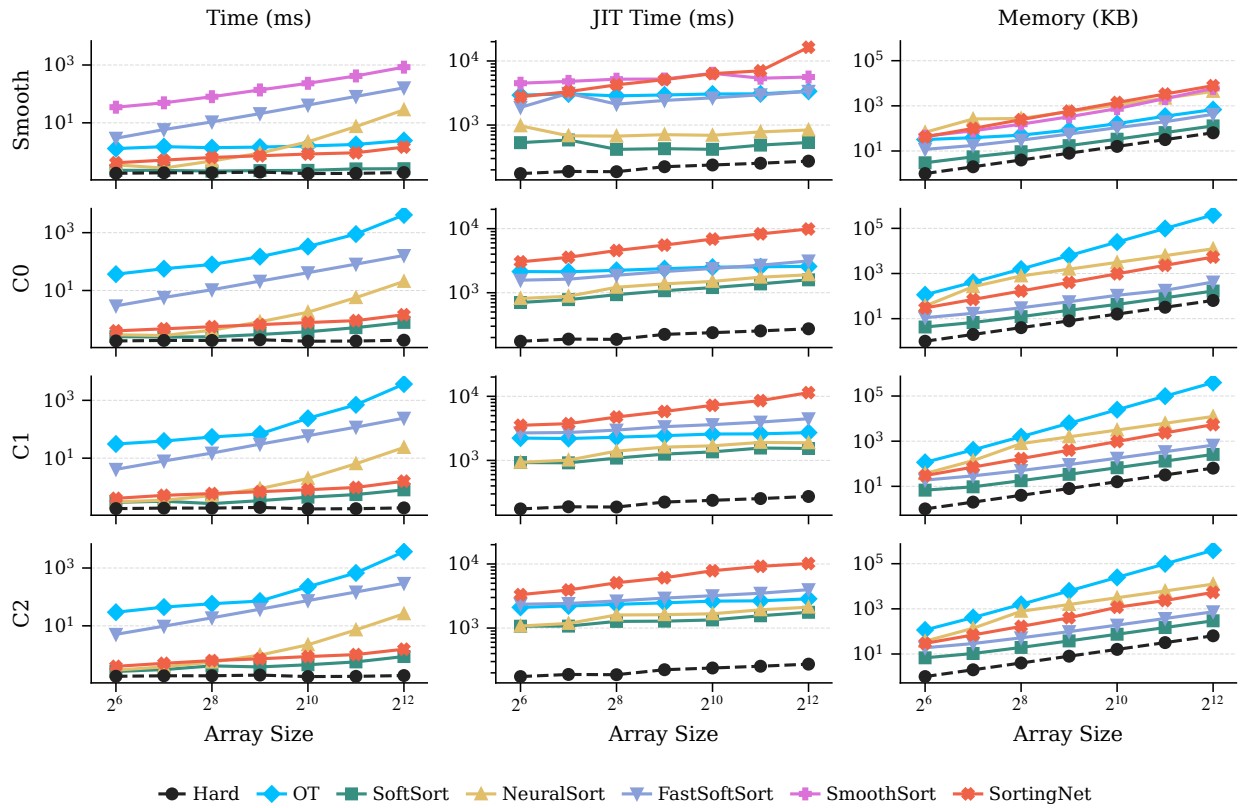

*Figure 19.* Benchmark results for `sj.max`.

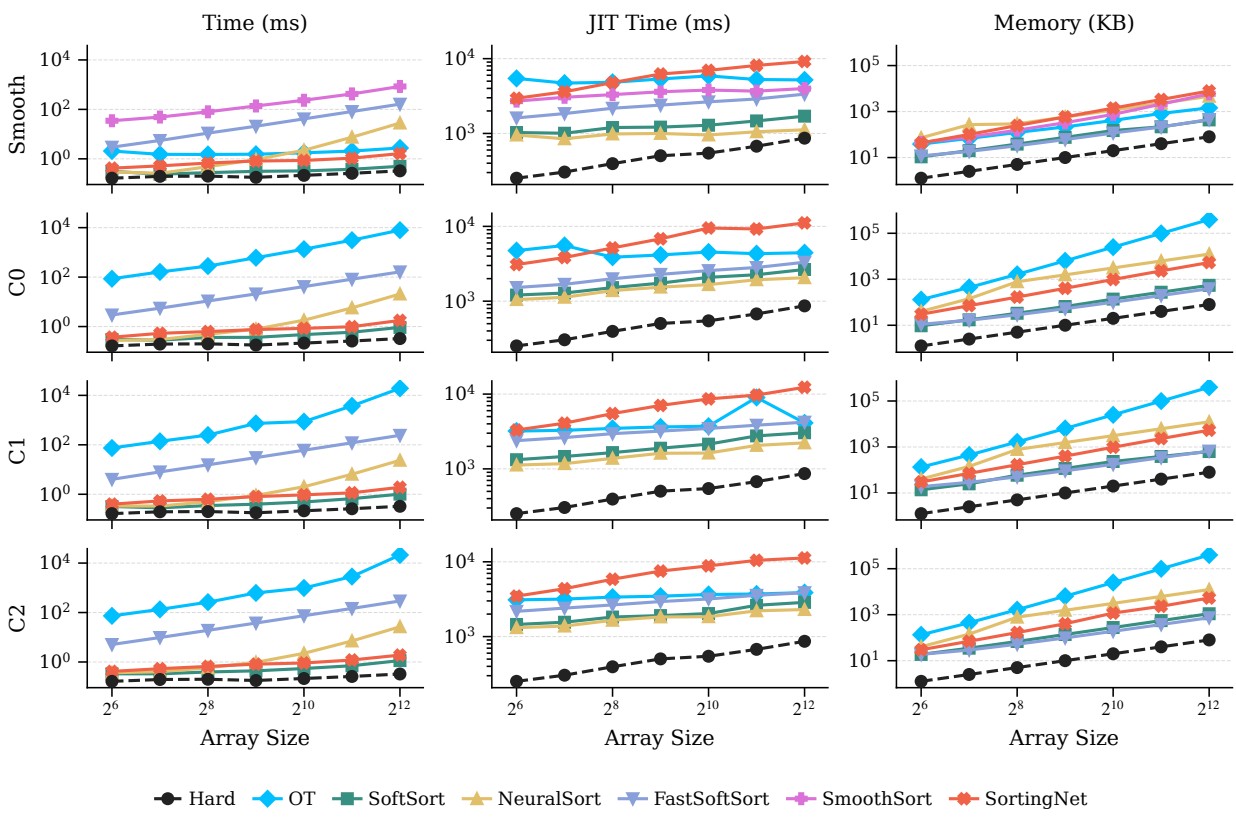

*Figure 20.* Benchmark results for `sj.top_k`.

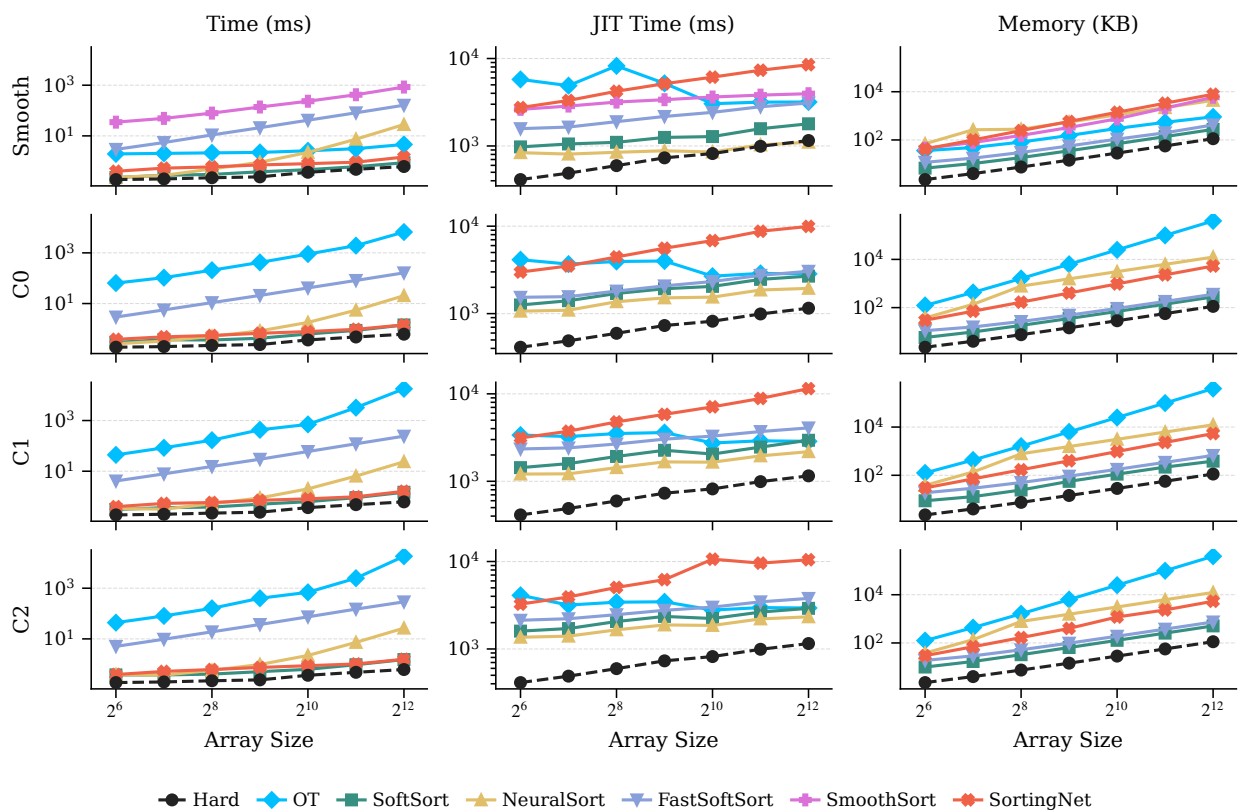

*Figure 21.* Benchmark results for `sj.median`.

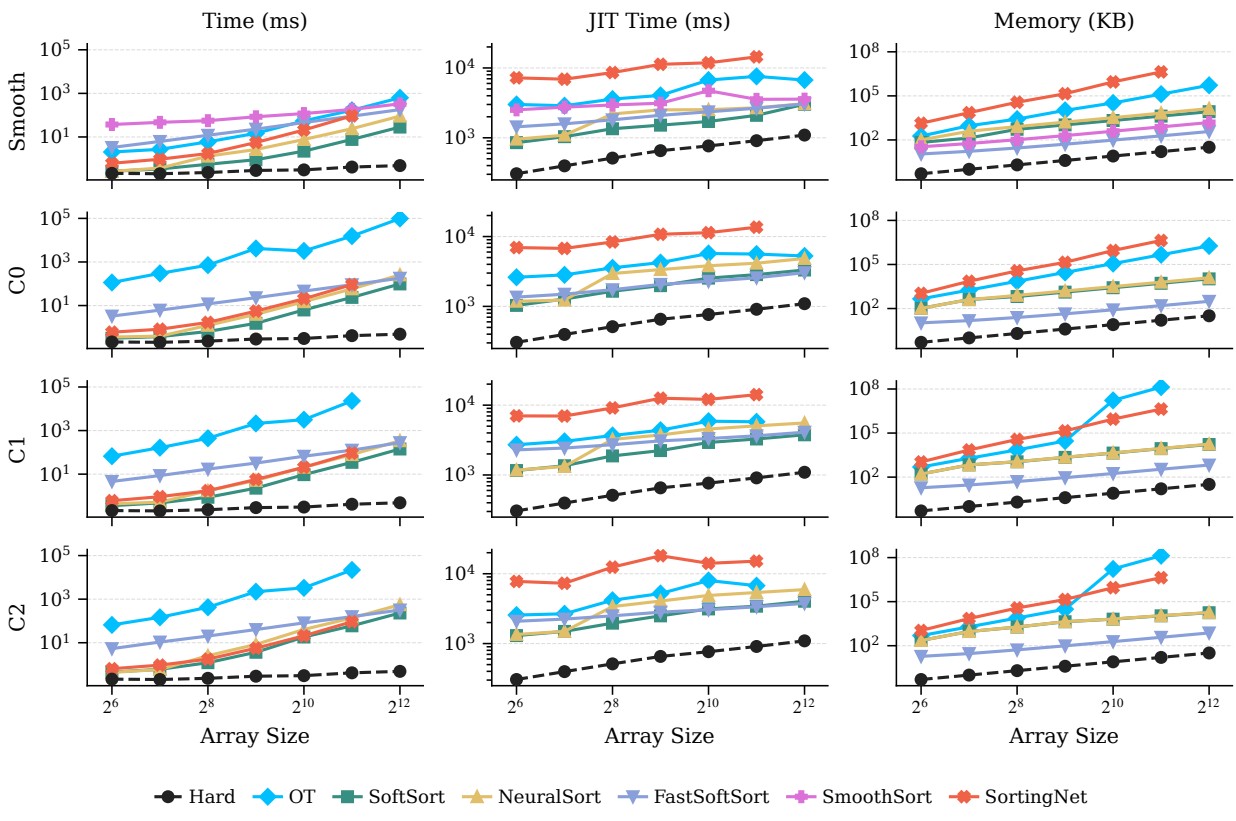

*Figure 22.* Benchmark results for `sj.rank`.

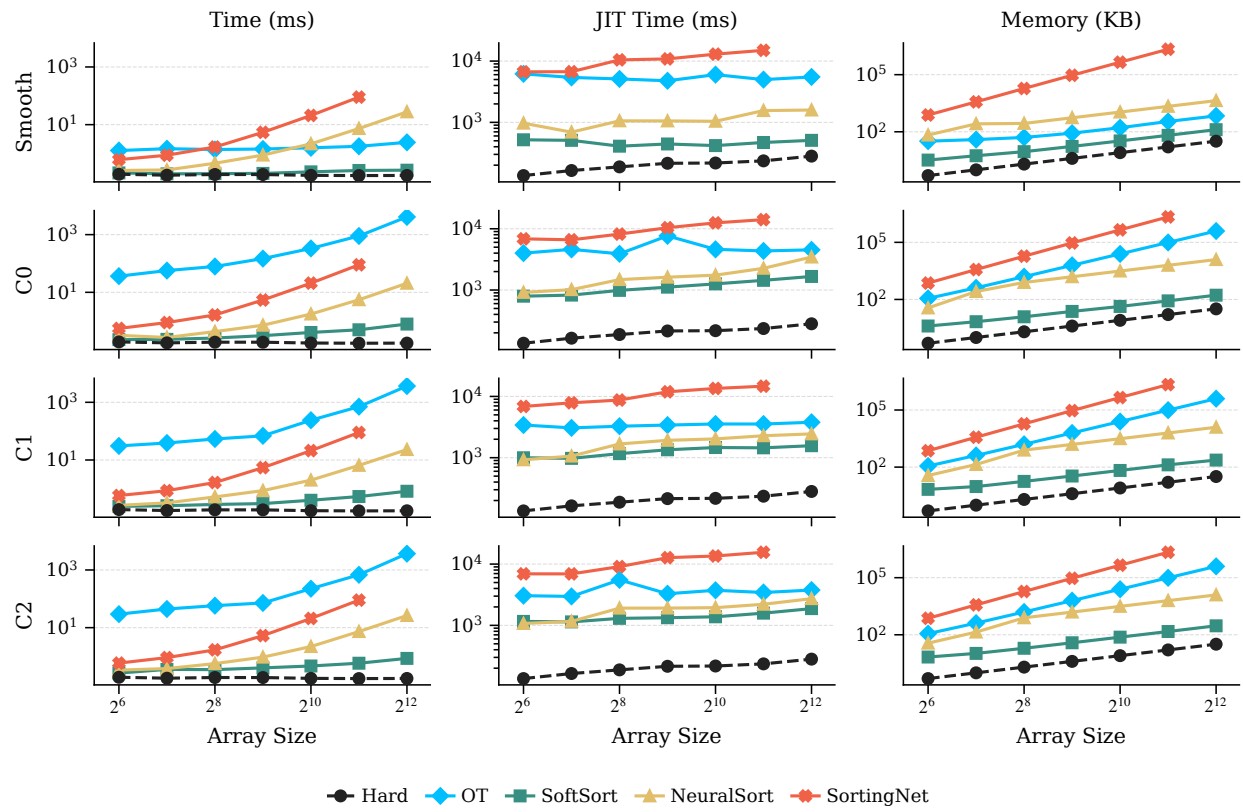

*Figure 23.* Benchmark results for `sj.argmax`.

# F. Choosing a Method and Mode

The preceding benchmark results reveal clear tradeoffs between the available methods and modes. This section provides practical guidance for choosing among them.

**Method selection.** For most use cases, we recommend **NeuralSort** or **sorting networks** as the default methods. Sorting networks are the fastest method and are highly parallelizable on GPUs (see Figure 8). NeuralSort is also fast with low memory consumption and fast JIT compilation in SoftJAX, and produces fully smooth surrogates in `smooth` mode (by using soft abs instead of the hard sort in SoftSort's cost matrix). Both methods support all operators, including soft index-producing functions like argsort and argmax.

When memory usage is the primary concern, **FastSoftSort** is the most memory-efficient method, as it avoids materializing the $n \times n$ permutation matrix entirely. However, FastSoftSort only produces sorted values and ranks, not soft indices (i. e., no argsort or argmax), and is not fully smooth due to its reliance on the hard sort for initialization.

For a more detailed comparison of all methods, we refer to the benchmark results in Figures 8 and 17 to 23.

**Mode selection.** For gradient-based optimization, we recommend `smooth` ($\mathcal{C}^\infty$) as the default mode. It produces the best-conditioned gradients and is compatible with higher-order differentiation (e. g., Hessian computation).

When a **bounded region of change** is desired, the piecewise modes `c1` and `c2` are good alternatives. These provide $\mathcal{C}^1$ and $\mathcal{C}^2$ smoothness respectively, with sparser gradients that are exactly zero outside the transition region. If Hessians are needed, `c2` should be preferred over `c1`.

The `c0` mode is generally **not recommended** for gradient-based optimization. It is only $\mathcal{C}^0$ (continuous but not differentiable everywhere), and the axiswise operators that rely on optimization-based projections (e. g., OT) can be numerically unstable in `c0` mode, because the L2 regularization induces sparse transport plans with discontinuous optimality conditions (Theorem 1).

**Softness parameter.** For elementwise operators, all modes share an effective transition width of approximately $10\tau$ (see Section D.3). For axiswise operators, the same $\tau$ value produces different effective softness across modes and problem sizes $n$. Users needing comparable behavior across modes or sizes can measure the normalized entropy $H(\mathbf{P})/\log n$ of the soft permutation matrix and calibrate $\tau$ accordingly. As a rule of thumb, smaller $\tau$ produces sharper approximations closer to the hard operator, at the cost of less informative gradients.

