# OpenReview forum: "SoftJAX & SoftTorch: Empowering Automatic Differentiation Libraries with Informative Gradients"
_ICML.cc/2026/Conference — ICML 2026 spotlight_

### Official Review · Reviewer_bkH4 · 2026-03-03

**Soundness:** 3
**Presentation:** 4
**Significance:** 3
**Originality:** 2
**Overall Recommendation:** 5
**Confidence:** 4

**Summary:**

***
After the response, I find all my questions have been answered, so I have raised my score

***

The paper describes a new software package (or rather two, for Jax and Torch), that collects and efficiently implements smooth (or soft) relaxations of many non-differentiable (or hard) operations. The paper is also a reasonably good introduction into the current methods and issues. It also introduces a few new such soft functions, and issues when composing soft functions.

**Compliance With Llm Reviewing Policy:**

Affirmed.

**Final Justification:**

After the response, I've raised my score because the paper is a clear, one-stop references implementation for end-to-end differentiable proxies for non-differentiable functions, it's well written, and fills a clear gap.

**Key Questions For Authors:**

Here are my main questions:

For the Element wise operators, you show how providing a soft implementation of the Heaviside function is all that is needed to derive soft approximations for many functions (round, abs, less_equal ..etc). But are these good approximations? I see no real justification or explanation as relating to the quality of the approximations.  To make my question precise, first consider the piece-wise polynomial-based sigmoidals in equation (6). You then offer three examples of odd polynomials. Why these polynomials? Are these the best minimax polynomial approximations? That is, are they the solution to
$$
\min_{p \in  \mathcal{P} } \max_{x \in [-\tau/2,\tau/2]} |H(x)-p(x)|
$$

If so, great, at least it’s clear they are the best polynomials in this $[-\tau/2, \tau/2]$ interval. Now how good of an approximation do you get when using $H_{\tau}(x)$ in eq (7-12)? Could you do much better if you instead derived the minimax polynomial approximation to these individual functions? There are even recent papers that derive the optimal minmax polynomial for the sign function, see [1] below. Can it be argued that using your approximation in (7) could result in the min-max polynomial of the sign function?

For your new Softsort, how does it compare to previous baselines such as Neuralsort? When would your Softsort be a better drop in replacement?
Is there an example use where guarding against the product rule pitfall when using the straight-through really helps?





[1] The Polar Express: Optimal Matrix Sign Methods and their Application to the Muon Algorithm
Download, Noah Amsel et al, arXiv:2505.16932, 2025


Minor points

Page 6, top of 2nd column, “For q-quantiles we slightly vary the approach of Cuturi et al. (2019)..” This will only make sense to readers who have read and remember Cuturi et al. (2019). Here you need to be more specific.
Equation (34), you need to define $\Pi_{\tau}$
Section B: “we soften *one* of the key subroutines”. Also this example is very nice to see how easy it is to drop and replace softjax.

**Limitations:**

This is difficult, since there are many ways to implement, and trade-off the soft surrogates for speed or being a good fit. But in any case, I don't think the authors have really discussed this.

**Strengths And Weaknesses:**

Collecting all of the soft/smooth relaxations in one place, with a unified interface, and GPU efficient implementation has a lot of value. Further strengths include

1. Pointing out this pitfall regarding the straight through operator when using the product rule
2. Proposing a novel SoftSort-style rank operator


But now, this brings me to my main criticism. How can the reader appreciate how useful are these two new contributions above? I don't see any experiment validating their use. I see in the appendix you do mention how to avoid the STE pitfall on your collision detection example, but you do not compare how turning on or off your fix for the product rule affects your example.

To some extent, this criticism extends to the other methods, except that, most all methods are known, and taken from papers where they have been validated to some degree. So in this case, you need only benchmark how efficient (assuming correct) your implementations are, which you have done for SoftSort, FastSoftSort, and the logical operators (in the appendix), which I think is good enough.

Furthermore, how different, or how does the implementation compare to the dispersed pieces of code made available online? For instance, I checked that one the authors you most reference, Mathieu Blondel, has some code for soft surrogates, such as
"smooth-ot" for smooth optimal transport.

---

> ### Author Rebuttal · Authors · 2026-03-31
>
> We thank the reviewer for the detailed review, and the time invested in helping us improve our work.
>
> **W1 and Q2:** *No experiment validating the STE pitfall and the dense variant (SmoothSort), as well as other methods.*
>
> We would like to stress that the main contribution is the library and the unified treatment in the paper. The STE pitfall is not a central contribution, but a practical observation. We will clarify its scope in the revision.
> Note that the paper presents the extreme case (both factors zero), but a single hard factor in a product is already sufficient to block gradients for all other factors. Any hard gate, mask, or indicator multiplied with a downstream computation will zero out the gradient for that computation when the gate is off. This situation is common in practice.
> That said, a concrete (admittedly slightly contrived) example of the exteme case is MuJoCo's contact force computation. The contact force depends on the impedance $d(r) \in [0,1]$, a function of signed distance $r$ that is zero when bodies are not touching ($r > 0$) (see https://mujoco.readthedocs.io/en/stable/modeling.html#impedance). The contact force decomposes depends on it through a stiffness term proportional to $d(r)^2$. To obtain pre-contact gradient signal without altering the forward simulation, one can use STE with a soft surrogate that extends $d$ to positive distances. Naively applying the STE to the softened $d(r)$ zeros out the stiffness gradient via the product rule for $r>0$. Applying STE to the square term preserves the stiffness gradient signal critical for discovering contact.
>
> Regarding the new soft sorting variant: since the submission we have heavily extended the benchmark suite, which showed it is significantly slower than the other methods in many cases. In the revised version we will present it as a more theoretical contribution on how to obtain smooth gradients using permutahedron projections (fastosoftsort always produces non-smooth behavior, even for log-KL regularization).
>
> Regarding evaluation of other methods, we have adapted a standard differentiable arg-sort experiment from the literature, and additionally, have added some better quick examples to the libraries on median regression, top-k feature selection, threshold filtering and logic rule classification. For the results, see the response to W2 of reviewer hTJG.
>
>
> **W2:** *How does the implementation compare to existing code like `smooth-ot`?*
>
> Existing implementations like `google-research/fast-soft-sort` and `smooth-ot` cover individual methods. Our libraries combine them under a single API with 40+ operators, multiple sorting/ranking methods, 4 regularization modes, and consistent softness parameterization, for both JAX and PyTorch.
> Moreover, there are some implementation differences.
> - For `fast-soft-sort`, the original codebase uses NumPy with a custom backward pass, so it cannot be JIT-compiled or vmapped. Our implementation is pure JAX, supporting native `jit` and `vmap`. We also extend the algorithm with closed-form block solvers for c1 and c2 modes, whereas the original only supports L2 and log-KL regularization.
> - For `smooth-ot`, that library is a NumPy/SciPy solver without autodiff support, so it cannot be used for gradient-based learning. Our OT implementation is fully differentiable via implicit differentiation, is GPU-accelerated, and supports non-entropic modes (c0, c1, c2) beyond entropic OT.
>
>
> **Q1:** *Why these polynomials? Are these the best minimax polynomial approximations?*
>
> The polynomials are Hermite interpolants chosen for their smoothness class, not for minimax optimality.
> We desire C^k regularity, which sets k+1 conditions on the polynomial at -1 and 1 and hence, there is a unique solution with a polynomial with degree n=2k+1, and our polynomials are exactly these. We will clarify this in the revision.
>
>
> **Q2:** *How does SmoothSort compare to NeuralSort? When does the STE fix help?*
>
> See W1 for both points. To summarize: SmoothSort provides a fully smooth permutahedron projection (unlike FastSoftSort which has discontinuities), but is significantly slower than NeuralSort and the other simplex-based methods in practice. The STE pitfall is relevant when hard functions interact multiplicatively; we will clarify its scope in the revision.
>
>
> **Minor points.** Thank you for these pointers, we will fix them in the revised version.
>
> **Limitations:** We have added an appendix section "Choosing a Method and Mode" (Appendix F) that provides practical guidance on selecting among methods, regularization modes, and softness values, with references to the substantially expanded benchmarking section.

---

> > ### Author Rebuttal · Reviewer_bkH4 · 2026-03-31
> >
> > Dear Authors,
> >
> > thank you for your answers, in particular I appreciate your transparency around STE pitfall and SmoothSort, in particular that Smoothsort is significantly slower than Neuralsort, and I understand and accept the main contribution as being a differentiable, jit-able, package for all such smooth approximations.
> >
> > As for the choice of polynomials below equation (6), ok, I understood they are chosen so that the resulting approximation is a $C^k$ continuous for some k.
> >
> > My other questions have also been addressed. After reading the other reviews, and given your response to my questions, I have decided to raise my score to 5.

---

### Official Review · Reviewer_CtUH · 2026-03-06

**Soundness:** 3
**Presentation:** 4
**Significance:** 4
**Originality:** 4
**Overall Recommendation:** 6
**Confidence:** 3

**Summary:**

The authors present two new Python libraries that extend either JAX or PyTorch with soft approximations to certain operations, which so far have had issues when naively applying automatic differentiation through them. Their motivation is to provide more *meaningful gradients* for optimization. To do so, the authors collect existing approaches for important operations, systematically derive new ones, and implement them as new libraries that allow for a (mostly) drop-in replacement for certain JAX or PyTorch primitives. The authors benchmark memory and runtime implications for different approaches to realize their relaxation and, in the appendix, provide a first case-study of making the mesh-to-mesh collision detection in Mujoco-XLA softly differentiable.

**Compliance With Llm Reviewing Policy:**

Affirmed.

**Final Justification:**

Since the authors promised to/have added unit tests, documentations, guides on choosing $\tau$, comparisons against non-soft versions, and achieving feature parity for both the JAX and PyTorch versions of their library, I decided to increase my score to 6.

**Key Questions For Authors:**

1. How do the soft approximations behave on higher-order derivatives (applications could be PINNs, 2nd order optimizer, optimizing deep implicit layers)? Do they work under nested autodiff calls? Could they even work under Taylor-mode autodiff?
2. Are all implementations safe under all JAX transformations? (i.e., JIT, vmap, transposition)?
3. How do the methods compare against their default operations regarding compute and memory (this would be an important baseline to add)?
4. Is the support for PyTorch as good as for JAX? Do the two libraries have feature parity? (According to the anonymous GitHub repo, the fast soft sorting is only available in JAX)
5. What is the intuition on the softenizer $\tau$ and what are reasonable value ranges for it?
6. Could the authors comment on weakness point (3)?

**Limitations:**

Only implicitly; I would encourage the authors to highlight performance penalties in comparison to default operations and (potentially unexpected) shape changes for all operations requiring sorting.

**Strengths And Weaknesses:**

The two presented libraries fill important gaps and can provide a reliable implementation for softening certain operations in modern machine learning frameworks. Moreover, I think these are strong points of the presented work:
1. The paper is well written, well structured, and overall has a polished feel.
2. The math is using the right level of rigor.
3. I appreciate the collection of the softening approaches that have popped up over the past years and have been scattered across publications. This paper (and the library) could thereby also serve as a central linking hub to this exciting field of research.
4. It is strong that they present libraries for both JAX and PyTorch.
5. The authors have added code to their publication via anonymous repositories.

While I do not see any major weaknesses with this manuscript, I do have a few smaller concerns:
1. For sorting-based approaches that produce matrices with distributions, it is not a direct drop-in replacement because the user likely does not expect to have an array returned with such higher shapes. While I agree that this is probably the only way to relax this operation, I would encourage the authors to accompany their published package with necessary documentation and examples to make this clear for an end user.
2. I hope the authors improve the documentation of their code upon release and extend it with unit tests. At the moment, the code is pretty bare-bones and could use more verbose variable names.
3. It could be my misunderstanding, but why do the Euclidean and the p-norm regularizers additionally have $\tau/2$ or $\tau/p$ pre-factors (line 269 right column). Based on this, the optimization objective in eq (24) would be quadratic in $\tau$. Is this intended?
4. When assessing the impact on memory and compute in figure 8 and figures 14 & 15, I would find it important to add the respective scaling behaviors of the non-soft default operations to truly assess the impact of the softening. In other words, I would be interested in seeing what is the impact of using the soft operation over the regular one.
5. Minor: Could the authors change the color mapping in their scaling experiments of fig 8 and fig 15. (Especially on printed paper,) the "OT" and the "FastSoftSort" entries are almost indistinguishable.
6. I feel there is a missing guide that provides intuition on how to choose the softenizer $\tau$ .

Overall, I have the impression this is a well-needed package that seems to straightforwardly extend two popular autodiff engines. I believe this is a very valuable contribution to the community and a suitable publication for ICML. Hopefully, the final code release is accompanied by good documentation to help users get started easily.

---

> ### Author Rebuttal · Authors · 2026-03-31
>
> We thank the reviewer for their thorough and constructive review and for the positive assessment.
>
> **W1 and W2:** *Add documentation, examples, and tests.*
>
> Yes we agree the documentation and testing was somewhat limited in the version we submitted. The submitted code actually already contains detailed docstrings for all public functions, formatted for MkDocs with usage examples, demo notebooks and cross-references. We have since added the full MkDocs site configuration and expanded the documentation with explicit examples showing how `argsort` returns a soft permutation matrix (SoftIndex type) and how to use it with `take_along_axis` for differentiable selection.
> We have also improved the code quality and variable naming.
> Additionally, both libraries now have an extensive test suite covering forward pass correctness, gradients and Hessians verified against finite differences, and JAX transform safety.
> Since ICML policy does not allow updating the code submission until the decision is made, these additions will appear in the camera-ready release.
>
> **W3 and Q6:** *Error in regularizers.*
>
> Thank you for catching this. The pre-factors were indeed written incorrectly, we have corrected this for the revised version.
>
> **W4 and Q3:** *Add non-soft baseline to benchmarks.*
>
> Agreed, we have added the "hard" baseline to all benchmark figures as a black dashed line (again, this will unfortunately only be visible in the camera-ready revision per ICML policy). The comparison shows that the simplex-based methods (SoftSort, NeuralSort) are within a small constant factor of the hard baseline in both runtime and JIT compilation time at moderate sizes, while memory overhead depends strongly on whether a full permutation matrix is produced.
> We have also since added a differentiable sorting network method [Petersen et al., ICML 2021], which is the fastest tested method at the cost of higher memory usage.
> We discuss these tradeoffs and give method recommendations in a new "Choosing a Method and Mode" appendix section (see W6).
>
> **W5:** *Fix colors in plot.*
>
> Thank you for the suggestion; we have modified the colors and added distinct marker shapes per method.
>
> **W6 and Q5:** *Missing guide on softness intuition.*
>
> Thanks for this suggestion. We have added an appendix section "Choosing a Method and Mode" that discusses method selection, mode selection, and softness calibration. For elementwise operators, all modes share an effective transition width due to cross-mode normalization (Section 3). For axiswise operators, the same tau produces different effective softness across modes and problem sizes. We suggest measuring the normalized entropy H(P)/log(n) of the soft permutation matrix to calibrate across settings.
>
> **Q1:** *How do the soft approximations behave on higher-order derivatives (applications could be PINNs, 2nd order optimizer, optimizing deep implicit layers)? Do they work under nested autodiff calls? Could they even work under Taylor-mode autodiff?*
>
> We have added over 1,100 tests in SoftJAX and 241 in SoftTorch for JAX transforms and higher-order derivatives, parametrized over all operator/method/mode combinations. All elementwise and comparison operators support correct Hessians across all modes. Axiswise operators do as well, except OT in c0 mode, where the L2-regularized transport plan has discontinuous optimality conditions that prevent stable implicit differentiation at second order.
> We have not yet tested Taylor-mode autodiff via `jax.experimental.jet`, but this would be worth exploring.
>
> **Q2:** *Are all implementations safe under all JAX transformations? (i.e., JIT, vmap, transposition)?*
>
> Yes. All operators pass under `jit`, `vmap`, and `jit(vmap)` composition for both forward and backward passes. This is verified in the test suite for softjax.
>
> **Q4:** *Is the support for PyTorch as good as for JAX? Do the two libraries have feature parity? (According to the anonymous GitHub repo, the fast soft sorting is only available in JAX)*
>
> Both libraries now support the same operators, methods, and modes, following the naming conventions and return types of their respective base frameworks. FastSoftSort in particular is now also available in SoftTorch as well, using Numba JIT compilation for the isotonic regression solver.

---

> > ### Author Rebuttal · Reviewer_CtUH · 2026-04-01
> >
> > I thank the authors for clarifying all my open points.
> >
> > This work is a great contribution for the ICML community!
> >
> > Since the authors promised to/have added unit tests, documentations, guides on choosing $\tau$, comparisons against non-soft versions, and achieving feature parity for both the JAX and PyTorch versions of their library, I decided to increase my score to 6.

---

### Official Review · Reviewer_BpPG · 2026-03-12

**Soundness:** 3
**Presentation:** 3
**Significance:** 3
**Originality:** 2
**Overall Recommendation:** 5
**Confidence:** 3

**Summary:**

Automatic differentiation frameworks are usually employed with hard tools for operations, such as ReLU, to enable faster computations. The proposed approach combines several softening approaches along with some new discussions (such as STE pitfall problem) in software packages as SoftJAX and SoftTorch. The packages are shared from an anonymous GitHub environment. The main possible novelty of the manuscript can be summarized in two folds: a) Unified package contains various softening approaches, b) handling the STE ptifall ( which can be defined as the appearance of the original functions in the backward pass due to the differentation). The authors provide a benchmark study demonstrating the performance gains of the proposed method over standard approaches in terms of CPU and memory complexity.

**Compliance With Llm Reviewing Policy:**

Affirmed.

**Final Justification:**

The authors clarified my concerns during the rebuttal process. I updated my score to 5.

**Key Questions For Authors:**

1. What will be the impact of softening approaches on the target ML problem accuracy?
2. Does the proposed software package provide a solution to the mentioned STE-pitfall problem?

**Limitations:**

The authors wrote a very generic impact statement.

**Strengths And Weaknesses:**

The paper presents a software package with benchmarks and anonymous code sharing. Providing anonymous access to the codes improves the reliability and reproducibility of the results. There is also a dedicated section discussing Benchmarks from the time and memory-complexity perspectives, which further strengthens the presentation of the provided software package. To handle the cases with zero gradients in hard-tool operations is a required topic in DL area. That's why the manuscript (and the provided software package) can be considered as a timely unifying attempt. However, the novelty claim is limited beyond the unification of softening approaches in the manuscript. On page 3, lines 135-160, the authors discussed that the possible problem may diminish the impact of softening. This problem may occur, especially when the product of functions appears in the gradient calculation. As stated by the authors in the same page lines 157-158, handling this issue can improve the usability of this software package in real problems. However, discussion on this part is very limited in the current version of the manuscript, and this part can be elaborated by the authors to improve the claim.

The discussion on the computational performance is already strong and provides some evidences for low-resource requirement of the proposed softening package. However, this issue should again be linked with the main purpose of the automatic differentiation. It means that an additional experiment showing the accuracy behaviour of a target machine learning problem will increase the probability of demand for the proposed software package. From an implementer's point of view, the impact of the softening operation on the accuracy of the target ML problem is as important as the computational performance.

---

> ### Author Rebuttal · Authors · 2026-03-31
>
> We thank the reviewer for their careful reading and for recognizing the software package as "a timely unifying attempt."
>
> **W1:** *"novelty claim is limited beyond the unification of softening approaches in the manuscript"*
>
> We kindly ask the reviewer to reconsider the utility/novelty of our work for the field of soft differentiable programming. The main manuscript sets up a unified framework including a discussion of the STE pitfall and the newly introduced smooth variant of fastsoftsort. The appendix adds clear connections between all elementwise oeprators as special cases of axiswise operators in C.3, discusses a novel way of computing gradients of axiswise operators in C.5 ("integration-like" gradient behavior), and introduces the entropy-regularized dual of the permutahedron projection in C.6.
> These contribution are in addition to providing a unified software library (including generalizing various previous methods to different regularizers and modes).
> Also note that in the revised version (which by ICML policy we cannot upload during the rebuttal), we also include a theorem that proves that for optimial transport and unit-simplex projection based methods, the regularizers we choose lead to the correct smoothness classes (e.g. entropic leads to smooth, different p-norm regularizers lead to continuous/differentiable/twice differentiable relaxations).
>
>
> **W2:** *Discussion on the STE pitfall is very limited and can be elaborated.*
>
> We will expand the STE pitfall discussion in the camera-ready version. The `st()` decorator in our library handles the pitfall automatically (see Q2 below). Note that even a single hard factor in a product blocks gradients for all other factors, so the issue is actually not limited to the extreme case presented in the paper.
> That said, a concrete (admittedly slightly contrived) example of the exteme case is MuJoCo's contact force computation. The contact force depends on the impedance $d(r) \in [0,1]$, a function of signed distance $r$ that is zero when bodies are not touching ($r > 0$) (see https://mujoco.readthedocs.io/en/stable/modeling.html#impedance). The contact force decomposes depends on it through a stiffness term proportional to $d(r)^2$. To obtain pre-contact gradient signal without altering the forward simulation, one can use STE with a soft surrogate that extends $d$ to positive distances. Naively applying the STE to the softened $d(r)$ zeros out the stiffness gradient via the product rule for $r>0$. Applying STE to the square term preserves the stiffness gradient signal critical for discovering contact.
>
>
> **W3:** *An additional experiment showing accuracy on a target ML problem is needed.*
>
> Thank you for this suggestion, shared by other reviewers. We have adapted a standard differentiable arg-sort experiment from the literature, and additionally, have added some better quick examples to the libraries on median regression, top-k feature selection, threshold filtering and logic rule classification. For the results, see the response to W2 of reviewer hTJG.
>
>
> **Q1:** *What will be the impact of softening approaches on the target ML problem accuracy?*
>
> As shown in various previous works and the additional experiments / quick examples, various problems in applied domains such as robotics or rendering *require* softening to make any progress on the ML problem at all. In other cases, softening allows enableing additional gradient streams ont he backward pass which can accelerate learning (simple example is using non-relu activation functions).
>
>
> **Q2:** *Does the proposed software package provide a solution to the mentioned STE-pitfall problem?*
>
> Yes. The `st()` decorator wraps any composite function so that STE is applied at the outer level, avoiding the product-rule pitfall.
> The `st()` decorator works on any user-defined function, not just library primitives. The case study in Appendix B shows this on collision detection in MuJoCo-XLA.
>
>
> **Limitations:** We have revised the impact statement to be more specific, discussing expected positive applications in robotics, simulation, and structured prediction, and shared risks with all AD frameworks. Since ICML policy does not allow updating the paper until the decision is made, this revision will appear in the camera-ready version.

---

> > ### Author Rebuttal · Reviewer_BpPG · 2026-04-03
> >
> > Thank you for clarifying all my concerns.  I would increase my score to 5.

---

### Official Review · Reviewer_hTJG · 2026-03-13

**Soundness:** 3
**Presentation:** 3
**Significance:** 3
**Originality:** 2
**Overall Recommendation:** 4
**Confidence:** 3

**Summary:**

The authors propose two open-source library SoftJax and SoftTorch for automatic differentiation of soft relaxations of hard operators, such as Relu and others. There are multiple implementations of soft variants of various hard operators including non-smooth elementwise functions, axiswise operators including sorting, top_k, and optimal transport.

**Compliance With Llm Reviewing Policy:**

Affirmed.

**Final Justification:**

The authors addressed my concerns.  Since the authors claim that their would provide test and examples, I increased my score from 3 to 4. Unfortunately, I can't increase it more, since they were not originally presented and can't be verified.

**Key Questions For Authors:**

see weaknesses

**Strengths And Weaknesses:**

The paper is dedicated to the two AD libraries. I find the libraries quite useful for the ML community. The paper is clear and well-written. There are a lot of useful soft implementation presented in the proposed libraries. However, for me, it seems that the libraries should be more wrapped up and supported as libraries.

Firstly, there are no tests to validate the correctness of the functions and the gradient calculation. Because of lack of tests, it is not clear for me, if it is possible for other users to enhance the library with new soft implementations.

Secondly, there are no proper use case examples inside of the networks or with calculations of the gradients. "quick-example" only shows calculations of the values not the gradient use case. Additionally, it would be helpful to compare the "soft" versions of some networks with "hard" to show the benefits of "soft" computations.

---

> ### Author Rebuttal · Authors · 2026-03-31
>
> We thank the reviewer for their review and the time invested in helping us improve our work.
>
> **W1:** *Firstly, there are no tests to validate the correctness of the functions and the gradient calculation. Because of lack of tests, it is not clear for me, if it is possible for other users to enhance the library with new soft implementations.*
>
> Since submission, we have fully packaged the libraries with a detailed MkDocs documentation, including introduction notebooks and example scripts, and upon acceptance will upload the project to PyPI to make it pip installable.
> Moreover, we have added an extensive test suite, where we verify that gradients match central finite differences across all operators, methods, and modes, and that hard mode reproduces native JAX/PyTorch operations exactly. The tests also check convergence of soft outputs to hard results as softness approaches zero, Hessians against finite differences, safety under `jit`, `vmap`, and their composition, and expected properties such as SoftIndex distributions summing to 1 and SoftBool values lying in [0,1].
> The testing framework is also very easily extendable, such that it will be easy for users to add new soft implementations.
> (Note that the ICML policy does not allow updating the code submission until the decision is made, but it will be included in the camera-ready release.)
>
>
> **W2:** *Secondly, there are no proper use case examples inside of the networks or with calculations of the gradients. "quick-example" only shows calculations of the values not the gradient use case. Additionally, it would be helpful to compare the "soft" versions of some networks with "hard" to show the benefits of "soft" computations.*
>
> We believe that the soft operators we collect have already been shown beneficial in the many papers that originally propose them.
> That said, we agree that the library should include more practical examples. We have therefore reimplemented a commonly used experiment used in (Grover et al., 2019; Cuturi et al., 2019; Blondel et al., 2020; Prillo et al., 2020; Petersen et al., 2021; 2022a).
> In this experiment, each sample is a sequence (of length n) of concatenated MNIST digits representing four-digit numbers. A CNN maps each image to a scalar score, and these scores are passed to a differentiable argsort operator. The model is trained against the ground-truth permutation using a cross-entropy loss and evaluated using exact-match and element-wise ranking accuracy (in brackets).
>
> The results match the results reported in (Prillo & Eisenschlos, 2020) with the addition of OT and Sorting Networks (additional method added since submission) performing similarly. Note that we also ran these experiments in "hard" mode (orginal JAX functions) which due to the zero-gradient arising from the jax.argsort operation resulted in no training.
>
> | smooth ($\boldsymbol{\tau=0.1}$) | $n=3$ | $n=5$ | n=7 | n=9 | n=15 |
> |---|:---:|:---:|:---:|:---:|:---:|
> | SoftSort | 93.0 (95.3) | 82.6 (92.2) | 69.5 (89.2) | 58.8 (87.3) | 24.0 (79.07) |
> | NeuralSort | 92.5 (94.9) | 81.7 (91.8) | 69.8 (89.4) | 58.6 (87.4) | 25.5 (79.7) |
> | Sorting Network | 92.6 (94.9) | 81.6 (91.8) | 67.5 (88.5) | 48.5 (84.0) | 16.6 (76.3) |
> | c2 ($\boldsymbol{\tau=0.1}$) | n=3 | n=5 | n=7 | n=9 | n=15 |
> | SoftSort | 88.0 (91.7) | 80.6 (91.5) | 66.4 (88.1) | 56.2 (86.8) | 30.2 (81.7) |
> | NeuralSort | 90.0 (93.2) | 78.2 (90.3) | 63.4 (87.1) | 49.2 (84.2) | 8.5 (69.7) |
> | Sorting Network | 90.3 (93.4) | 79.7 (90.9) | 65.2 (87.7) | 45.9 (82.9) | 13.1 (74.0) |
> | OT | 87.6 (91.5) | 84.1 (93.0) | 73.2 (90.7) | 58.0 (87.2) | 27.1 (80.7) |
>
> Furthermore, we added another usecase "Soft Rasterization with SoftJAX" which implements a minimal soft rasterizer based on the ideas of SoftRas (Liu et al., NeurIPS 2019). The example optimizes 2D triangle vertex positions to match a target image rendering, showing that the hard rasterizer produces zero gradients at overlapping edges (optimization fails) while the soft rasterizer (SoftJAX) provides informative gradients (optimization converges).
>
> Finally, we have also added additional "quick examples" to the documentation, including median regression (regress median of 3D vector onto a target), top-k feature selection (linear regression model perturbed by noise, select 3 most influential features), threshold filtering (given number array learn a threshold using sj.greater such that the selected entries sum to two), and logic rule classification (a feature vector is equipped with a true label if any of its entries are in the range [0.3,0.6], learn this rule using logic gates with STE).
> Note that for all these examples we include both the soft and hard results, with "hard" mode being either unable to make progress as the gradient is zero or showing large parameter oscillations leading to suboptimal convergence (in the case of median regression).

---

> > ### Author Rebuttal · Reviewer_hTJG · 2026-04-03
> >
> > Dear Authors,
> > thank you for extensive response. I appreciate the provided answers. Since the authors claim that their would provide test and examples, I would increase my score to 4. Unfortunately, I can't increase it more, since they were not originally presented and can't be verified.
> > Best wishes.

---

### Decision · Program_Chairs · 2026-04-30

**Decision:**

Accept (spotlight)

**Comment:**

The reviewers agreeed positively on this paper presenting SoftJAX and SoftTorch, two unified libraries of soft differentiable drop-in replacements for hard JAX/PyTorch primitives. Initial concerns around missing tests, limited examples, and absent hard baselines were credibly addressed in the rebuttal. The theoretical contributions beyond software packaging add sufficient novelty. I strongly recommend acceptance.